# Oxonium ion scanning mass spectrometry for large-scale plasma glycoproteomics

Matthew E. H. White[1], Ludwig R. Sinn [2], D. Marc Jones[3,4], Joost de Folter [5], Simran Kaur Aulakh[1], Ziyue Wang[2], Helen R. Flynn [6], Lynn Krüger[7,8], Pinkus Tober-Lau[9], Vadim Demichev[1,2], Florian Kurth [9], Michael Mülleder [10], Véronique Blanchard[7,8], Christoph B. Messner[1,11,13] ✉ & Markus Ralser [1,2,12,13] ✉

Protein glycosylation, a complex and heterogeneous post-translational modification that is frequently dysregulated in disease, has been difficult to analyse at scale. Here we report a data-independent acquisition technique for the large-scale mass-spectrometric quantification of glycopeptides in plasma samples. The technique, which we named 'OxoScan-MS', identifies oxonium ions as glycopeptide fragments and exploits a sliding-quadrupole dimension to generate comprehensive and untargeted oxonium ion maps of precursor masses assigned to fragment ions from non-enriched plasma samples. By applying OxoScan-MS to quantify 1,002 glycopeptide features in the plasma glycoproteomes from patients with COVID-19 and healthy controls, we found that severe COVID-19 induces differential glycosylation in IgA, haptoglobin, transferrin and other disease-relevant plasma glycoproteins. OxoScan-MS may allow for the quantitative mapping of glycoproteomes at the scale of hundreds to thousands of samples.

The proteomes of liquid biopsies and peripheral body fluids, in particular blood plasma or serum, are an emerging source of biomarkers, bearing potential for novel diagnostic, prognostic and predictive applications[1,2]. The plasma proteome contains important nutrient response proteins, coagulation factors and components of the immune system, whose concentration and activity reflect the physiological condition of the individual and which are therefore important for precision medicine[3–5]. Technologies facilitating the quantification of the plasma proteome in large sample series, using mass spectrometry[2] or with the affinity-reagent-based Olink[6] and SomaScan[7] platforms, have opened exciting avenues to better link genetic diversity and disease phenotypes at the epidemiological scale[8]. However, the activity and function of proteins depends not only on their abundance but also on post-translational modifications. These mediate protein–protein and protein–small molecule interactions, processes that themselves depend on whether a protein is modified[9]. Consequently, abundance measurements alone capture only part of the human physiology represented by the plasma proteome, creating a need to develop methods

[1]Molecular Biology of Metabolism Laboratory, The Francis Crick Institute, London, UK. [2]Department of Biochemistry, Charité – Universitätsmedizin Berlin, Corporate Member of Freie Universität Berlin and Humboldt-Universität zu Berlin, Berlin, Germany. [3]Bioinformatics and Computational Biology Laboratory, The Francis Crick Institute, London, UK. [4]Department of Basic and Clinical Neuroscience, Maurice Wohl Clinical Neuroscience Institute, London, UK. [5]Software Engineering and Artificial Intelligence Technology Platform, The Francis Crick Institute, London, UK. [6]Mass Spectrometry Proteomics Science Technology Platform, The Francis Crick Institute, London, UK. [7]Institute of Diagnostic Laboratory Medicine, Charité – Universitätsmedizin Berlin Corporate Member of Freie Universität Berlin and Humboldt-Universität zu Berlin, Berlin, Germany. [8]Department of Human Medicine, Medical School Berlin, Berlin, Germany. [9]Department of Infectious Diseases and Critical Care Medicine, Charité – Universitätsmedizin Berlin Corporate Member of Freie Universität Berlin and Humboldt-Universität zu Berlin, Berlin, Germany. [10]Core Facility High-throughput Mass Spectrometry, Charité – Universitätsmedizin Berlin Corporate Member of Freie Universität Berlin and Humboldt-Universität zu Berlin, Berlin, Germany. [11]Precision Proteomic Center, Swiss Institute of Allergy and Asthma Research (SIAF), University of Zurich, Davos, Switzerland. [12]Max Planck Institute for Molecular Genetics, Berlin, Germany. [13]These authors contributed equally: Christoph B. Messner, Markus Ralser. ✉e-mail: christoph.messner@siaf.uzh.ch; markus.ralser@charite.de

that can address post-translational modifications and proteoforms at cohort scale.

Glycoproteomics is considered an important reservoir for biomarker discovery. Protein glycosylation is abundant and diverse in plasma, and altered glycosylation has been observed in response to a variety of disease states, for example, prostate-specific antigen in prostate cancer and alpha-1-acid glycoprotein in sepsis[10–13]. Therefore, there is an increasing demand for approaches that allow the sensitive and quantitative profiling of blood plasma, where protein glycosylation plays a vital role in regulating the structure and function of both soluble and cell-surface proteins[14]. Liquid chromatography–mass spectrometry-based (LC–MS) proteomic technologies are widely applied in the identification and quantification of post-translational modifications in cell-derived and tissue-derived samples[9,15–20]. Furthermore, through advances in sample preparation and novel data-acquisition strategies, MS-based technologies have also reached a level of robustness and throughput for large-scale high-throughput investigations that involve the measurement of thousands of samples[5,21–24].

However, the study of intact glycopeptides at scale still presents a number of analytical challenges. A large proportion of glycoproteins have multiple glycosylation sites (macroheterogeneity), at each of which there is a large range of possible glycan structures (microheterogeneity). The abundance of a given glycoprotein therefore comprises various individual glycoforms at lower respective concentrations, necessitating a highly sensitive analytical approach[25,26]. Furthermore, co-elution of unmodified peptides reduces sensitivity via ion suppression, and for data-dependent acquisition, by reducing the time spent by the instrument specifically sampling glycopeptides[27]. These effects are compounded by the poorer ionization efficiency of glycopeptides relative to their unmodified counterparts[28]. A number of glycoprotein/glycopeptide enrichment and analysis strategies have been developed to minimize the challenges of intact glycopeptide analysis[29,30]. These reach excellent depth on individual samples but have increased cost and handling time, and create potential batch effects, which limit their application on large cohort studies. Data-independent acquisition (DIA) methods, such as sequential window acquisition of all theoretical mass spectra (SWATH-MS), have been increasingly applied in the analysis of large proteomic sample series[31–35]. In glycoproteomics, DIA approaches have been applied to assess glycosite occupancy of enzymatically deglycosylated peptides[36–39], and more recently, facilitated the post-acquisition analysis of intact glycopeptides, either by targeted extraction of abundant Y-type (intact peptide with glycan fragments of various sizes) ions[40–44] or by searching against spectral libraries[18,45–47]. Both data-dependent acquisition (DDA) and DIA approaches yield remarkable depth in comparative analyses and in generating spectral libraries, generally using collisional-based dissociation (either higher-collisional dissociation (HCD) or collision-induced dissociation (CID)) and/or electron-based fragmentation techniques[47–49]. MS-based technologies have been further applied to quantify oxonium ions—small singly-charged fragment ions ubiquitously found in glycopeptide CID/HCD tandem mass spectrum (MS/MS) spectra[50–52] in biotherapeutics and purified glycoproteins, as well as in complex biofluids[40,43,53–61].

Here we present a glycoproteomic screening approach for high-throughput studies. In contrast to previous workflows, we take a two-step approach that separates glycopeptide quantification from sequence assignment. Specifically, in a fast screening step, we exploit the sensitive detection and quantification of oxonium ions diagnostic for individual glycopeptide features and combine it with a scanning quadrupole dimension, as introduced with Scanning SWATH[21], to assign precursor masses to quantified oxonium ions. The information obtained from the scanning dimension facilitates the matching of precursor and MS/MS information between OxoScan-glycoproteomics and DDA-glycoproteomics data for identification of the glycopeptides in the second step.

We demonstrate the application of OxoScan-MS using micro-flow chromatography by identifying 30 IgG glycoforms without predefined compositional knowledge, and further validate glycopeptide signal specificity and quantitative performance in tryptic digests of human plasma and serum. Moreover, we applied OxoScan-MS to generate a plasma glycoproteome for a cohort of 30 hospitalized COVID-19 (coronavirus disease 2019) patients and 15 healthy controls, in technical triplicates. On clinical citrate plasma samples, our approach quantified >1,000 glycopeptide features in just 19 min of active chromatographic separation across 164 samples, measured in just 3 d of instrument time. We selected a subset of quantitatively interesting glycopeptide features as potential glyco-biomarkers from the COVID-19 cohort and utilized an orthogonal acquisition approach (higher-collisional dissociation with oxonium ion-dependent triggering of electron-transfer dissociation fragmentation (HCD-pd-ETD)) to perform glycopeptide identification. Critically, our method captures quantitative biological variation in a plasma cohort. Follow-up analysis of glycopeptide features-of-interest and integration with protein-level data by targeted mass spectrometry identified potential biomarkers and differential glycan regulation with increasing COVID-19 disease severity. Thus, OxoScan-MS facilitates glycoproteomics on neat plasma at large scale, and we report its use for the untargeted cohort-level plasma glycoproteomic analysis of severe COVID-19.

## Scanning quadrupole allows for untargeted glycopeptide profiling

We previously described a DIA-based scanning quadrupole acquisition method, Scanning SWATH, in which a scanning quadrupole (Q1) facilitates assignment of precursor masses by time-dependent fragment ion detection in a DIA-MS experiment[21]. In OxoScan-MS, the scanning dimension allows the extraction of a 'Q1 profile' for fragment ions as the precursor enters and exits the sliding Q1 isolation window, centred on the precursor $m/z$. We demonstrate that selectively extracting Q1 profiles of oxonium ions, which are produced when glycans fragment under CID conditions[50–52], allows detection of glycopeptide precursors, even in the presence of co-eluting unmodified peptides (Fig. 1a,b). By overlaying Q1 traces with MS1 spectra, accurate masses can be assigned (Fig. 1c). As extracted ion chromatograms show glycopeptide elution in the chromatographic dimension (Fig. 1d), selectively extracting oxonium ion chromatograms across the entire precursor range generates a two-dimensional (2D) matrix of glycopeptide signals, even in complex samples containing mostly unmodified peptides (Fig. 1e). Not only does this remove the need for predefined knowledge of glycopeptide constituents and the biases associated with an empirical spectral library, but it also allows relative quantification between samples.

To test the validity of this principle, we first profiled IgG subclasses 1, 2 and 4, purified from human blood serum[62]. By extracting chromatograms of commonly identified oxonium ions across the acquired precursor range, an 'oxonium ion map' visually identified >30 features corresponding to the IgG glycopeptides (Fig. 1f and Extended Data Fig. 1a). It is worth noting that features represent unique retention time–precursor $m/z$ coordinates and are not unambiguously identified glycopeptides at the point of detection. Matching MS1 features to previously reported MS1 signals of glycopeptides (from matrix-assisted laser desorption ionization-time of flight mass spectrometry (MALDI-TOF-MS)[62] and nanoLC–MS/MS[63]) was used for the identification of 30 of these glycopeptide features (Supplementary Table 1). Moreover, we observed well-documented and reproducible retention time shifts for the glycopeptides of each IgG subclass, recapitulating known behaviour of both different peptide sequences between IgG subclasses and different glycans with reverse-phase separations (Extended Data Fig. 1b)[64,65].

Recent studies have shown the utility of Y-type fragment ions for quantification and generation of site-specific glycopeptide

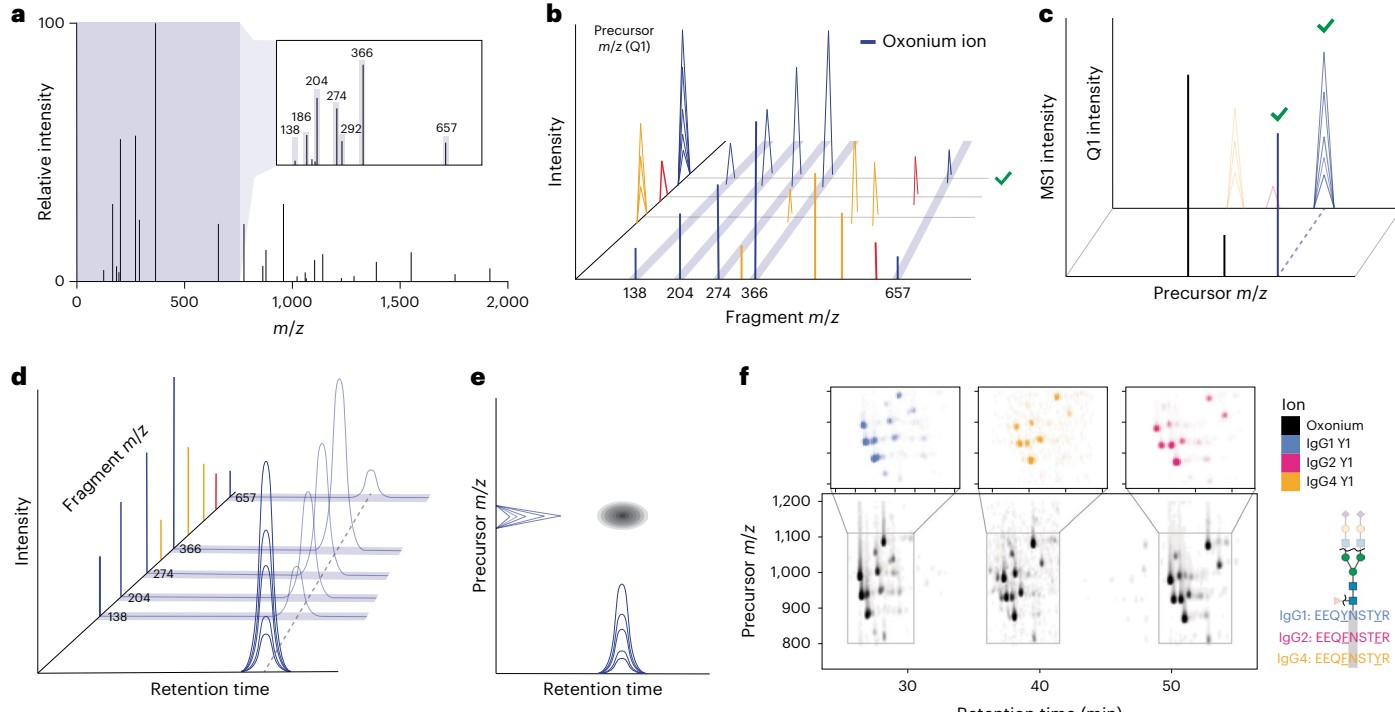

**Fig. 1 | OxoScan-MS exploits a scanning quadrupole for selective glycopeptide profiling by precursor assignment of glycan-specific ions. a**, Representative MS/MS spectrum from a glycopeptide fragmented under CID conditions, with the low-mass oxonium ion region highlighted in purple. The oxonium ions arising from fragmentation of HexNAc (138.05, 186.08, 204.09), Neu5Ac (274.09, 292.10), HexNAc-Hex (366.14) and HexNAc-Hex-Neu5Ac (657.24) ions are highlighted in the inset spectrum. These ions are very commonly detected upon fragmentation of glycopeptides by CID. **b**, Time-dependence of the scanning quadrupole within a single cycle gives a 'Q1 profile' of each fragment ion entering and exiting the sliding precursor isolation window, which is centred around the precursor mass. Blue signals denote oxonium ions, red and yellow denote co-eluting peptide precursors, which do not produce oxonium ions. Because oxonium ion masses are both specific to glycopeptides but largely ubiquitous across glycopeptide identities, their extraction can distinguish a glycopeptide from a co-eluting unmodified peptide. A further in-depth explanation of the scanning quadrupole concept and Q1 traces can be found in ref. 21. **c**, Glycopeptide precursors can be identified by overlaying oxonium ion Q1 profiles on MS1 spectra. Oxonium ion Q1 elution peaks are glycopeptide-specific,

as co-eluting unmodified peptides do not give rise to oxonium ion fragments. MS1 peaks with overlaid oxonium ion traces can therefore be identified and localized as glycopeptides, as shown by the green tick. **d**, XICs depict elution of a glycopeptide in the chromatographic time dimension by oxonium ion signals. Such XICs can be extracted for any fragment ion, but oxonium ion signals specifically denote the elution of a glycopeptide at a given retention time. **e**, Each glycopeptide feature, defined as a glycopeptide in a specific charge state, can be localized to a unique retention time–precursor *m/z* coordinate, with peak height (*z*-dimension in the shown plot) proportional to peak signal. **f**, Oxonium ion map of IgG 1, 2 and 4 glycopeptides with IgG peptide sequences coloured by subclass. Bottom panel shows the sum of oxonium ion intensities, where each cluster of spots corresponds to the glycopeptides of each IgG subclass at the conserved *N*-glycan site and each spot is a specific glycopeptide. For ease of interpretation, intensities for respective subclasses have been scaled separately. Top panels show the Y1 (peptide + GlcNAc) ions extracted and plotted for each IgG subclass, respectively. As Y1 ions are not ubiquitous to all glycopeptides but also depend on the peptide sequence, glycopeptides of different IgG subclasses can be distinguished.

information in DIA analysis[40,42,66]. On the basis of these observations, we developed a rolling collision energy scheme, such that the MS/MS spectra of each glycopeptide feature also contain useful Y-type fragments for targeted re-analysis. Although these spectra cannot yet be processed with currently available glycoproteomic search engines, we found that highly abundant fragments of peptides with 1–5 attached sugar molecules (the remainder of the glycans being preferentially fragmented over the peptide backbone) allow identification of features from the same peptide. Indeed, we find that Y1 (peptide + HexNAc) fragments in particular, when calculated in silico[40] and extracted in DIA-NN[34], overlay on their respective oxonium ion features, facilitating the distinction of glycopeptides from different IgG subclasses by their respective peptide sequences (Fig. 1f, top panels). This highlights a key advantage of OxoScan-MS: each run acts as a digital archive of the glycoproteome of a sample. Consequently, OxoScan-MS leverages the advantages of both a precursor ion scan and SWATH-MS in a single run for untargeted quantification of all glycopeptide features with oxonium ions above the limit of detection.

## Quantification of over 1,100 glycopeptide features in neat plasma

We next tested the performance of our method on human plasma. As a large proportion of plasma proteins are glycosylated, we expected to generate considerably more complex data than that obtained from purified IgG[67]. Analysis of a plasma sample prepared using a semi-automated high-throughput sample preparation pipeline[5] with OxoScan-MS (Fig. 2a) produced complex oxonium ion maps with hundreds of visible glycopeptide features (Fig. 2b). To confirm glycopeptide specificity of oxonium ion signals, we treated the sample with a cocktail of glycosidases (Protein Deglycosylation Mix II, New England Biolabs), which enzymatically cleave most glycan classes from proteins, leaving predominantly deglycosylated and non-glycosylated peptides. The glycosidase treatment results in a 99% reduction in oxonium ion signal intensity, illustrating the specificity of oxonium ion detection in OxoScan-MS for glycopeptides (Fig. 2c, bottom panels).

To extend this approach for automated and quantitative analysis of oxonium ion profiles, we applied a persistent homology-based[68] algorithm for 2D peak-calling and quantification. For each peak extending

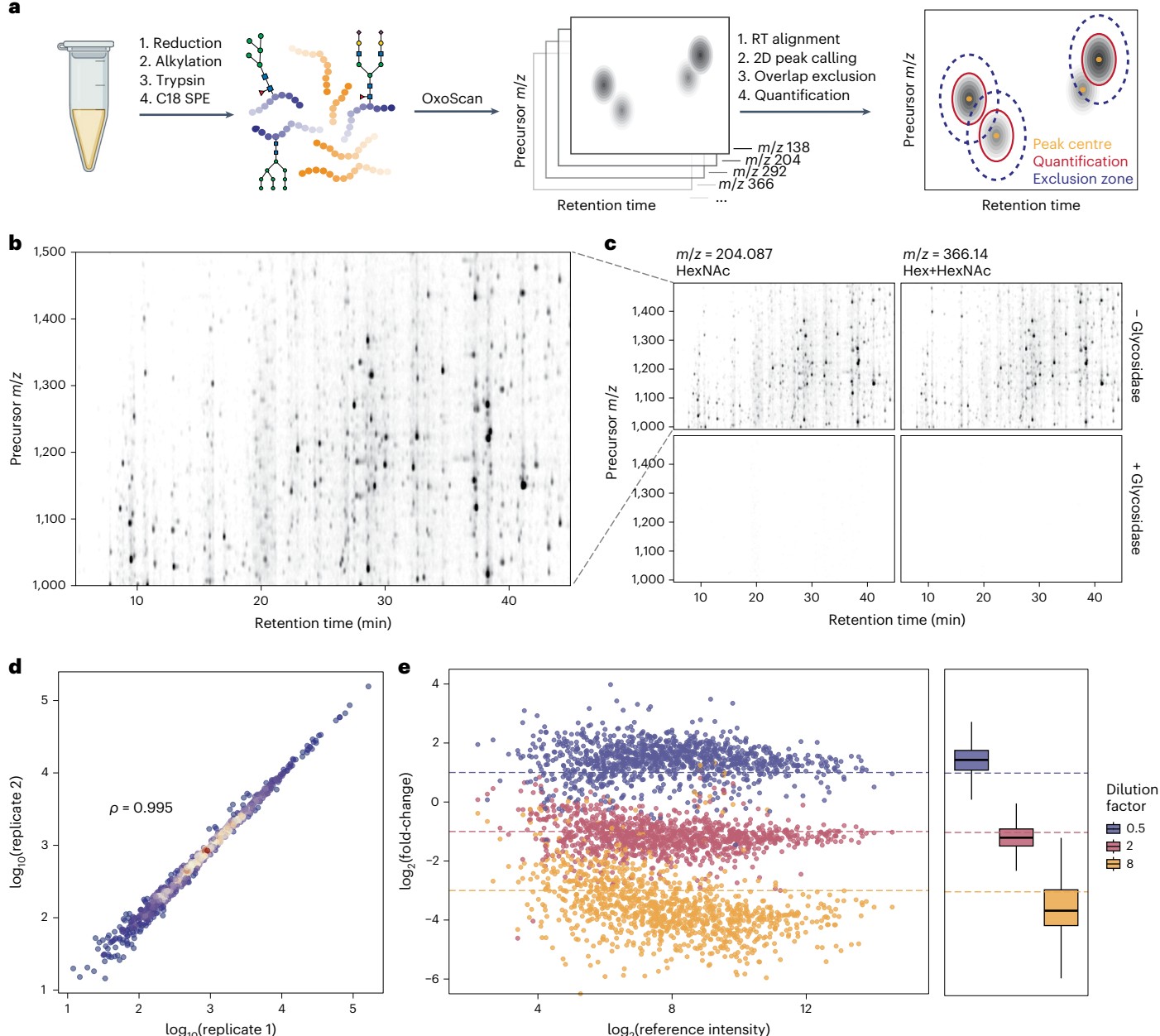

**Fig. 2 | Oxonium ion maps generate a specific and quantitative glycoproteome from the analysis of neat human plasma. a**, Oxonium ion profiling workflow, starting with the generation of oxonium ion maps from unenriched tryptic digests of serum/plasma (glyco)proteins and computational analysis. **b**, An oxonium ion map of human plasma tryptic digest, extracted for the $m/z = 204.09$ (HexNAc) oxonium ion. Each spot represents a glycopeptide in a specific charge state. **c**, Oxonium ion maps for two common oxonium ions present in tryptic digests of human plasma, with and without treatment with a mix of glycosidase enzymes (bottom and top panels, respectively). Peak intensity is proportional to opacity, and all panels are scaled to the maximum

peak intensity across the experiment. **d**, Comparison of intensities between two injections of human plasma tryptic digests. Spearman correlation coefficient was calculated on the basis of glycopeptide feature intensities ($n = 1,006$). **e**, $\log_2$(fold-change) plotted for serum glycopeptide features from spiking in 3 different concentrations into an *E. coli* tryptic digest ($n = 819$). Fold-changes were calculated to a reference dilution factor of 1 and theoretical $\log_2$(fold-change) values expected for each dilution factor are plotted as dashed lines. The box-and-whisker plots display 25th, 50th (median) and 75th percentiles in boxes; whiskers display upper/lower limits of data (excluding outliers, not plotted). Figure 2a created with BioRender.com.

into the intensity ($z$) dimension in an oxonium ion map, a 'persistence' score is computed, representing the vertical distance between peak maximum and the point where it merges into an adjacent higher peak. Theoretically, a peak resembling a 2D Gaussian function would have a persistence value equivalent to its height, whereas the persistence value of a peak shoulder would equate to the distance from its apex to the minimum point between the shoulder and the peak (Extended Data Fig. 1d). To facilitate comparison of multiple samples, we implemented

retention time alignment using dynamic time-warping[69]. Upon alignment, peaks are called and ranked by their persistence value. To prevent duplicate calling of a single peak, an exclusion criterion ('exclusion ellipse') can be set, within which the centre of another peak with a lower persistence value cannot be called. Quantification is then performed by summing all points in a customizable 'quantification ellipse' around each peak maximum. To make this analysis approach widely applicable and customizable, all Python functions and standalone notebooks

with analysis parameters and requirements are made freely available (https://github.com/ehwmatt/OxoScan-MS).

On neat human plasma tryptic digests, this pipeline identified >1,100 glycopeptide features (corresponding to a glycopeptide in a specific charge state) spanning over four orders of magnitude in abundance within just 19 min of chromatographic separation. Importantly, oxonium ion maps are generated separately for each oxonium ion extracted and show high overlap (Extended Data Fig. 1c) but are summed for all subsequent analyses. The quantities resulting from the 2D peak integration show high reproducibility between replicate injections of a plasma sample (Spearman $\rho$ = 0.994, Fig. 2d). We further confirmed quantitative performance by spiking a tryptic serum digest into a background of [13]C-labelled *E. coli* proteome, maintaining constant total protein content and varying the serum:*E. coli* proteome ratio. Peaks originating from plasma glycopeptide features were isolated by removal of any putative glycopeptide feature observed in a 100% *E. coli* sample. Observed fold-changes in each dilution compared to a reference sample showed agreement with theoretical fold-changes, indicating that differential abundance of glycopeptide features is captured by the OxoScan-MS workflow (Fig. 2e).

We further re-extracted less ubiquitously reported but highly clinically relevant oxonium ions (HexNAc-HexNAc, *m/z* 407.165; HexNAc-Hex-Fuc, *m/z* 512.197; HexNAc-Hex-Fuc-Neu5Ac, *m/z* 803.293) in a human plasma sample. Although of lower abundance, features for each oxonium ion are clearly visible on an oxonium ion map (Extended Data Fig. 2a) and even show overlay on ubiquitous oxonium ion peaks, as would be expected for glycopeptide-derived fragment ions (Extended Data Fig. 2b).

## The quantitative plasma glycoproteome of severe COVID-19

To test the applicability of OxoScan-MS for cohort studies, we analysed the plasma glycoproteome of a severity-balanced cohort of 30 patients hospitalized due to COVID-19 as well as 15 healthy controls[21]. Disease severity among patients was assessed according to the WHO (World Health Organization) ordinal scale for clinical improvement, ranging from grade 3 (hospitalized, not requiring supplemental oxygen) to grade 7 (requiring invasive mechanical ventilation and additional organ support, Fig. 3a). The study protocol and plasma sampling strategies of this cohort has been previously described[5,21]. We utilized micro-flow chromatography with a 19 min active gradient and scanned a precursor range optimized for glycopeptides (800–1,400 *m/z*, Extended Data Fig. 3a). Including blanks and quality-control (QC) samples, a total of 164 glycoproteomic samples were measured in ~3 d of instrument time (Fig. 3b). Applying our open-source analysis pipeline to the cohort detected 1,102 unique glycopeptide features across all samples, >90% (1,002) of which were consistently quantified across all clinical samples (see Methods for details). To assess quantitative reproducibility of the oxonium ion signatures identified, a coefficient of variation (c.v.) was calculated for each feature within the triplicate measurements of each sample. Repeated analysis of a pooled plasma sample ('mass spectrometer QC') and nine replicates of a commercial plasma standard sample (Tebu Bio) prepared alongside the clinical samples ('sample preparation QC') showed reproducibility across the batch measurements, with median c.v.s of 14% and 20%, respectively. Importantly, the changes observed in clinical samples (median c.v. = 44%) were much higher than this technical variation, indicating that our method detects biological differences (Fig. 3c). The dynamic range of quantified features spans over four orders of magnitude (Fig. 3d). Some 230 glycopeptide features were found to be significantly changing in response to severe acute respiratory syndrome coronavirus 2 (SARS-CoV-2) infection (Extended Data Fig. 3d, $\log_2$(fold-change) > 1, adjusted *P* < 0.05, Benjamini–Hochberg multiple testing correction). Consistent with the differential expression analysis, principal component analysis (PCA) and hierarchical clustering show that glycoproteomic profiles correctly clustered the majority of healthy and COVID patients (Fig. 3e,f), indicating differential glycopeptide abundances with increasing COVID-19 disease severity. For three COVID-19 patients, we observed clustering with healthy controls, one of which is explained by very mild disease. It is worth noting, however, that we observed this on both the protein level and the glycopeptide level[5,70].

As a next step, we sought to identify and validate glycopeptide features significantly changing with COVID-19 disease severity by analysing plasma pools of healthy and critically ill individuals by HCD-pd-ETD on an Orbitrap Eclipse (Thermo Fisher) (Fig. 4a). Recent studies have shown that glycoproteomic assignment can vary substantially with the analysis software and settings[71], so we performed glycopeptide identification with both Byonic[72] (Protein Metrics) and MSFragger-Glyco[73], and further filtering post-processing for assignment quality (DDA data processing in Methods). It is worth noting that both Byonic and MSFragger provide assignment of glycan compositions but do not inform on linkage-specific or structure-specific glycan characteristics. As such, the glycan identity assigned to a given glycopeptide feature reflects the monosaccharide composition, as opposed to specific structural assignment. While Byonic assigned a greater number of MS/MS spectra to glycopeptides than MSFragger-Glyco (2,433 vs 608 peptide-spectrum-matches (PSMs)), 82% of MSFragger-Glyco assignments were also shared in Byonic. To increase confidence, we kept only those assignments shared between both Byonic and MSFragger-Glyco, and mapped them to candidate precursor masses obtained by OxoScan-MS. We then performed detailed inspection for 22 out of 167 putative matches (see Methods) by high-resolution precursor ion matching (Fig. 4b), retention time agreement (Extended Data Fig. 3c), comparison of respective DDA-window and narrow-window DIA-derived MS/MS spectra (Fig. 4c and Extended Data Fig. 4), and validation of precise quantification ellipses (Fig. 4d). Among those validated glycopeptides, we identified distinct differences in glycopeptide abundances between healthy patients and increasing COVID-19 severity across a number of disease-relevant proteins, including haptoglobin, alpha-2-HS-glycoprotein, immunoglobulin A, transferrin and alpha-1-acid glycoprotein (Fig. 5a and Extended Data Fig. 5).

To confirm this quantification, we re-prepared the plasma cohort and analysed the samples by high-resolution multiple reaction monitoring (MRM-HR) on a ZenoTOF 7600 instrument (Sciex). Indeed, MS/MS spectra from MRM-HR and OxoScan-MS showed excellent agreement (Fig. 5b) and despite being prepared in a separate laboratory and measured on a different LC–MS platform, we observed similar quantitative changes across the cohort for the majority (17/22) of monitored glycopeptides (Fig. 5c). Furthermore, we observed that quantifying glycopeptide features by the sum of oxonium ion intensities agreed excellently with using glycopeptide-specific Y-type ions for quantification (Fig. 5d), further demonstrating that oxonium ions are a viable source of quantitative glycoproteomic information.

A change in specific glycopeptide abundance could be caused by regulation of relative glycan composition, site occupancy and/or a change in total protein abundance. To measure protein abundance changes in parallel, we further monitored unmodified peptides from the identified glycosylated proteins (termed 'adjacent' peptides) within the same MRM-HR run (Extended Data Fig. 7). Normalizing each glycopeptide to the aggregate intensity of adjacent peptides showed examples of glycopeptide changes explained simply by changes in protein abundance, notably for serotransferrin (TF) (N630, N4H5S2) and haptoglobin (HP) (N241, N4H5S2). Interestingly, while the abundance change of the TF glycopeptide (N630, N4H5S2) did not significantly deviate from the trend in protein abundance, the abundance of its non-glycosylated N630-containing peptide declined more sharply than that of the adjacent peptides (Extended Data Fig. 6a, c), potentially suggesting a change to an alternative post-translational modification occurring on this peptide[74]. We further identified several cases where the observed glycopeptide changes are significantly different from

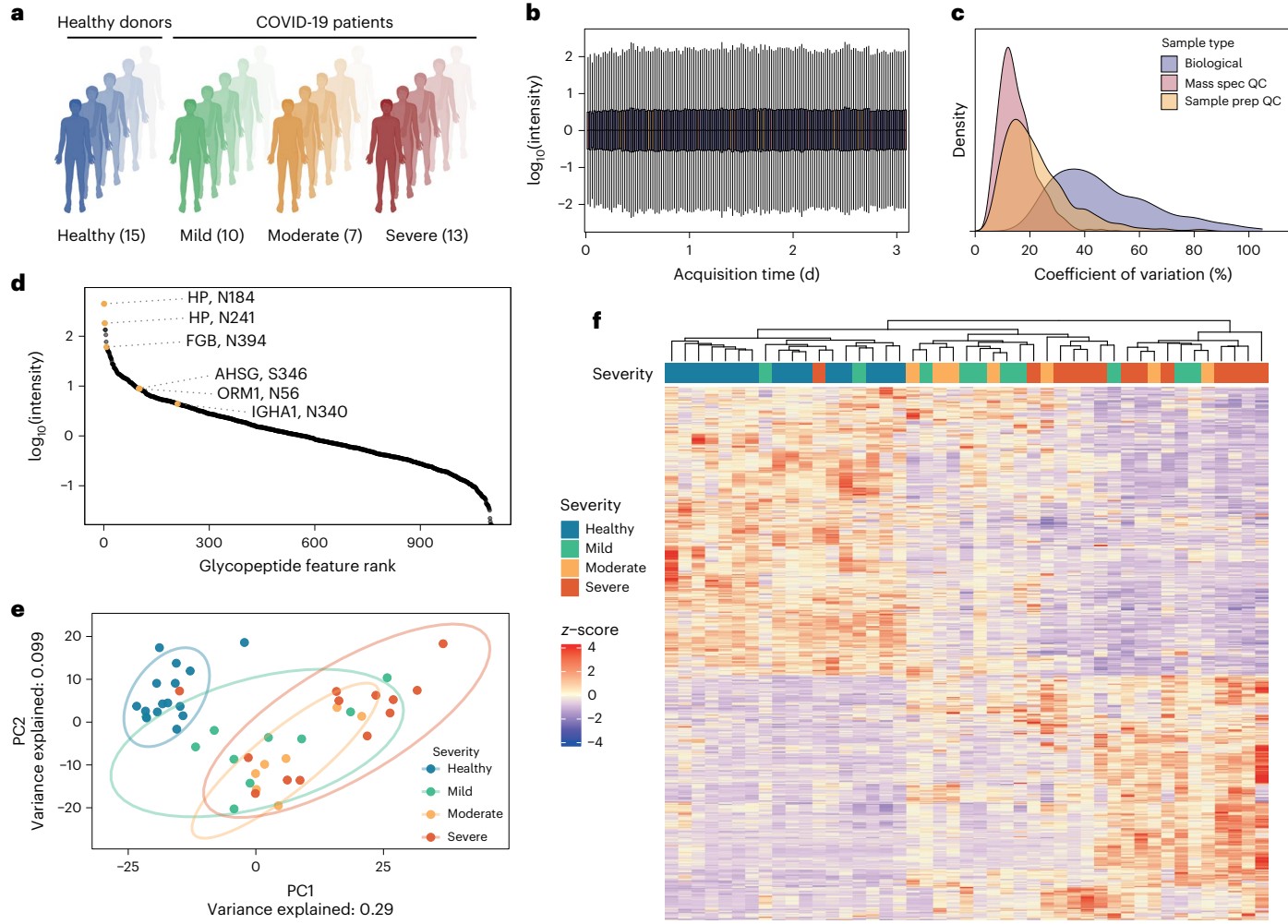

**Fig. 3 | Oxonium ion profiling allows robust and reproducible plasma glycoproteomics in a COVID-19 inpatient cohort. a**, COVID-19 inpatient cohort, comprising 30 patients hospitalized due to PCR-confirmed SARS-CoV-2 infection and 15 healthy controls. COVID-19 patients were distributed across different disease severities, ranging from mild (WHO 3), moderate (WHO 4, 5) to severe (WHO 6, 7) COVID-19. **b**, Total sample intensities across the MS measurement batch following median normalization; outliers are not plotted. Boxplot colours are the same as shown in panel **c**. **c**, Technical and biological variation across cohort measurements, indicated by distributions of c.v. values for glycopeptide features in repeat injections (mass spec QC, $n = 10$), commercial plasma (Tebu Bio) prepared in parallel with samples (sample prep QC, $n = 9$) and patient samples ($n = 3$ for each of 45 participants). **d**, Median intensity of glycopeptide features in a pooled sample, showing quantification spanning more than four orders of magnitude. Matched glycopeptide features are highlighted and labelled with their gene name and glycosite. **e**, PCA of all consistently detected features ($n = 1,002$) separates healthy and COVID-19 patients in PC1. The proportion of variation accounted for by each axis is shown in axis labels. **f**, Heat map and hierarchical clustering of differentially expressed glycopeptide features (calculated using the limma R package, |$\log_2$(fold-change)| > 1, adjusted $P < 0.05$) between COVID-19 patients and controls. Figure 3a created with BioRender.com.

the protein-level regulation. For example, *N*-glycans on both alpha-1-acid glycoprotein (ORM1) (N56, N4H5S2) and immunoglobulin A heavy constant A1/2 (IGHA1;IGHA2) (N144/131, N5H3) as well as an *O*-glycan on alpha-2-HS-glycoprotein (AHSG) (S346, N1H1S1) show an increase above protein-level changes as COVID-19 severity increases ($P < 0.01$, Kendall trend test, Fig. 5e and Extended Data Fig. 6c). These results demonstrate that glycoproteomics studies can detect both glycan-specific and, indirectly, protein-specific changes in clinical plasma cohorts and further reinforce the potential of clinical glycoproteomics in delivering disease-specific biomarkers that go beyond protein abundance measurements.

## Discussion

Recent studies have attributed high potential for the identification of next-generation glyco-biomarkers and predictive signatures[75–77], but due to the complexity of protein glycosylation, large-scale analysis of plasma and serum glycosylation remains a major challenge. Here

we present OxoScan-MS and demonstrate robust and reproducible quantification of over 1,000 glycopeptide features in neat plasma, with a total run-time per sample of less than 30 min and no requirement for glycopeptide enrichment. OxoScan-MS operates by scanning for and quantifying diagnostic oxonium ions, followed by targeted glycopeptide feature identification. OxoScan-MS is hence not a replacement for current glycoproteomic techniques; rather, it is a complementary method for fast, quantitative and cost-effective screening of large sample series. In contrast to DDA-based glycopeptide approaches where the co-elution of unmodified peptides reduces the time spent analysing glycopeptides specifically, OxoScan-MS samples glycopeptides independently of co-eluting unmodified peptides; it is therefore compatible with samples prepared for protein-level analyses, combining the advantages of a precursor ion scan with SWATH-MS to provide a digital snapshot of the glycoproteome. OxoScan-MS is specifically designed for the glycoproteomic profiling of hundreds to thousands of samples prepared for conventional MS-based proteomics.

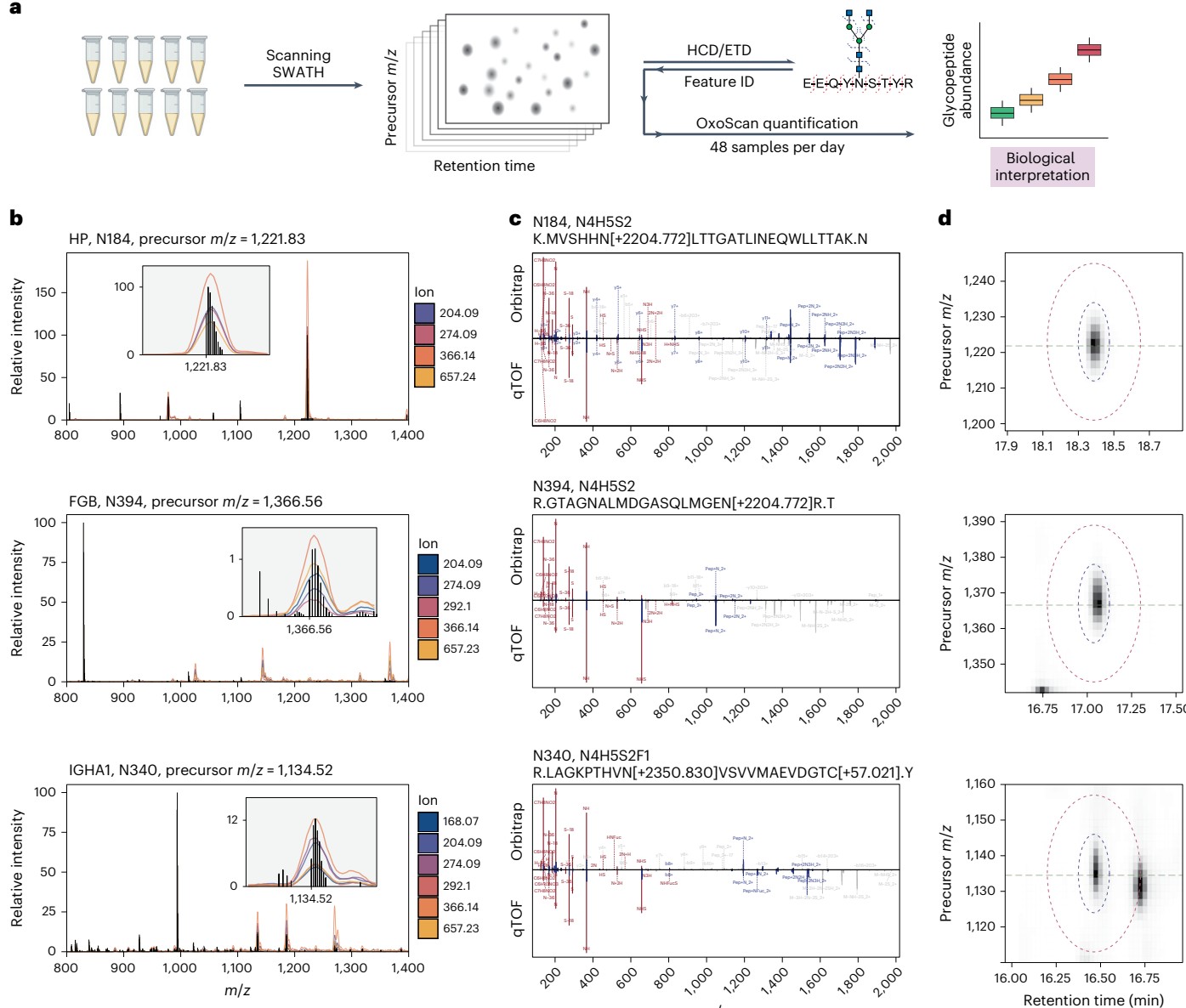

**Fig. 4 | Precursor assignment from the MS1 scanning dimension and subsequent MS/MS matching allow identification of candidate biomarker glycopeptides. a**, Plasma samples are measured using OxoScan-MS to generate oxonium ion maps, and glycopeptide features are identified with complementary fragmentation and database searching. OxoScan-MS then allows quantification of identified features across cohorts with >100s of samples. **b**, MS1 spectrum of tryptic plasma digest with Q1 profiles of oxonium ions overlaid. Oxonium ion traces localize glycopeptide precursor ions even in the presence of co-eluting unmodified peptides of significantly higher abundance. Inset shows zoomed-in oxonium ion traces with precursor *m/z* labelled on the *x* axis. Q1 profiles were acquired with a 2 *m/z* scanning window. Top: haptoglobin *N*-glycopeptide (Asn184). Middle: fibrinogen beta chain *N*-glycopeptide (Asn394). Bottom:

immunoglobulin A *N*-glycopeptide (Asn340). **c**, Comparison of DDA (HCD) and DIA (CID) MS/MS spectra for respective glycopeptide precursors. Fragment assignments are taken from analysis of DDA data in Byonic (with a tolerance of 5 ppm for DDA and 20 ppm for DIA). Fragments observed in both DDA and DIA spectra (also matched to within 20 ppm) are shown in blue and oxonium ions are shown in red. All non-matched assignments are shown in grey. Respective panels show the same glycopeptides as in **b**. **d**, Oxonium ion elution profiles in both precursor *m/z* and RT space for respective glycopeptide precursors. Blue and red ellipses represent the quantification and exclusion regions, respectively, and the horizontal line indicates accurate (TOF) precursor *m/z*. Panels show the same glycopeptides as in **b** and **c**. Figure 4a created with BioRender.com.

We applied OxoScan-MS to study the plasma glycoproteome in response to SARS-CoV-2 infection, measuring a severity-balanced clinical inpatient cohort in triplicate (164 samples in total) in just 3 d of instrument time. From the glycopeptide features measured, 230 were differentially abundant between healthy and severely affected patients. We then selected 22 features and determined their peptide identity and glycan composition using conventional glycoproteomic approaches. We found altered glycopeptide abundances among proteins important

in COVID-19, including haptoglobin, transferrin and immunoglobulin A (IgA). Furthermore, by integrating protein-level and glycopeptide-level analyses, we identified glycan-specific regulation dependent on COVID-19 severity, most notably for IgA, alpha-2-HS-glycoprotein (AHSG) and alpha-1-acid glycoprotein (ORM1). Reassuringly, ORM1, IgA and AHSG are indicators of COVID-19 disease severity[78,79] at the protein level, hence our results associated their differential glycosylation to severe COVID-19. Altogether, these results demonstrate disease-specific

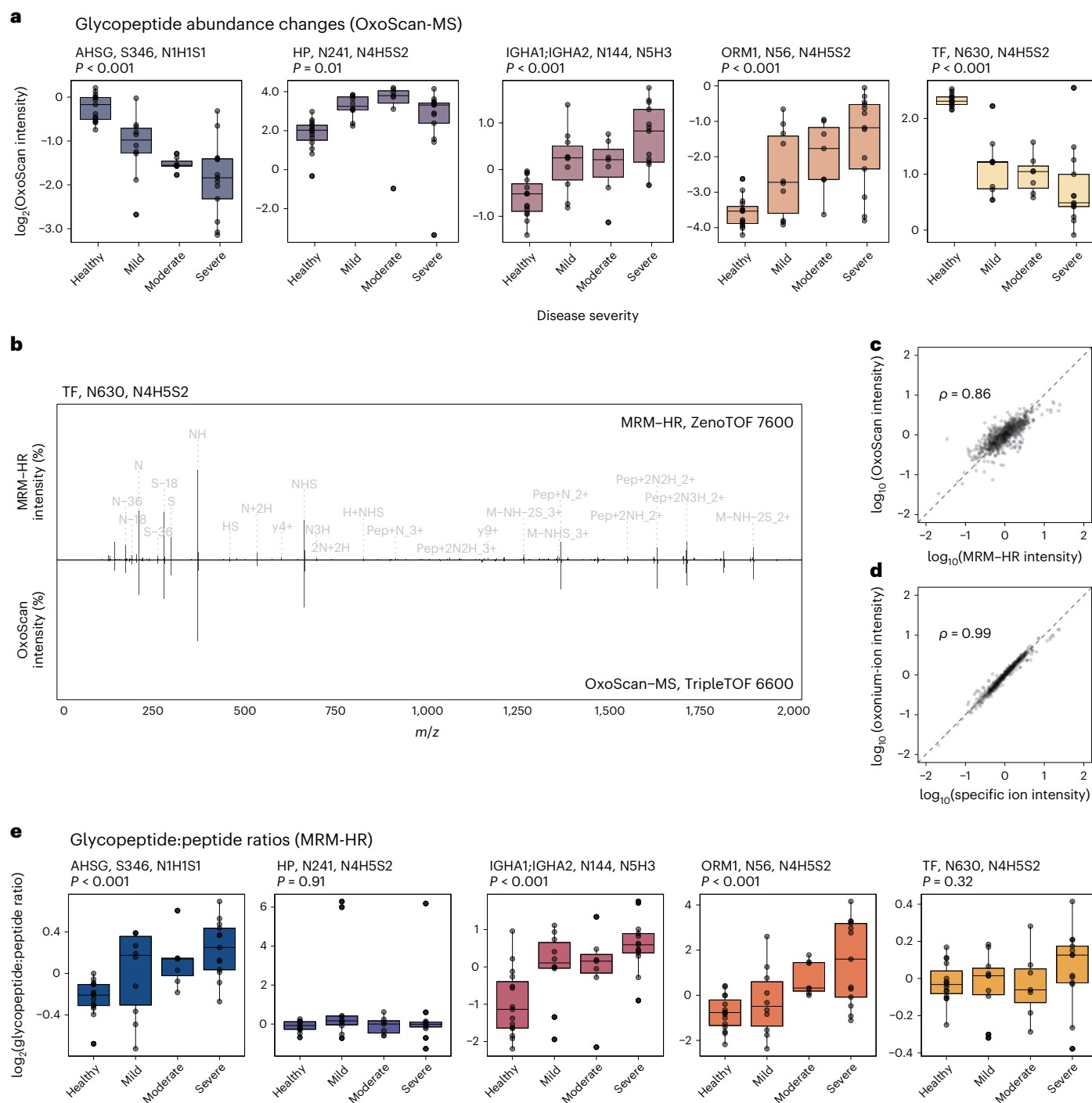

**Fig. 5 | OxoScan-MS identifies differential abundance of intact glycopeptides with COVID-19 disease severity.** Detection of site-specific regulation in the plasma glycoproteome of SARS-CoV-2 patients and healthy controls, first by OxoScan-MS and separately validated by MRM-HR in a second laboratory. **a**, OxoScan-MS intensities for five glycopeptides across a clinical COVID-19 cohort, demonstrating robust differential abundance of glycopeptides with disease severity. Significance was calculated using the Kendall–Tau test for the Theil–Sen trend estimator and adjusted for multiple testing according to the Benjamini–Hochberg FDR approach[99]. Boxplots display 25th, 50th (median) and 75th percentiles; whiskers display upper/lower limits of data. **b**, Representative back-to-back spectra from MRM-HR and OxoScan-MS fragment spectra, showing high overlap between identified fragment ions across different instruments and acquisition methods. Annotated peaks are shared between both MRM-HR and OxoScan-MS. Peak labelling was only displayed above a minimum base peak intensity of 2% for clarity. **c**, Correlation of OxoScan-MS intensities and MRM-HR intensities for validated glycopeptide targets show excellent agreement. The Spearman correlation coefficient was calculated using all validated glycopeptide features ($n = 17$). **d**, Oxonium ion intensities and glycopeptide-specific ions (Y-type) show excellent agreement. Spearman correlation coefficient was calculated using the sum intensities of oxonium ions ($m/z$ 138.055, 186.076, 204.087, 274.092, 292.103, 366.139, 657.235) and the 5 highest intensity specific ions identified in Skyline for all validated glycopeptide features ($n = 17$). **e**, Boxplots showing intensity ratios of each glycopeptide, normalized to adjacent non-glycosylated peptides from the same protein measured in the same MRM-HR run. At least 2 non-glycosylated precursors were used for normalization in each case (see Methods). For IGHA1;IGHA2, peptides shared between both subclasses were used for normalization, although no significant difference was seen between subclasses (Extended Data Fig. 6b). Significance and boxplot information as in **a**.

glycopeptide changes and the potential of glycoproteomics-based approaches for clinical biomarker development.

It is worth noting that in line with the tools used for glycopeptide identification, we report glycan compositional changes, as opposed to detailed structural or linkage information, which represents an established challenge in glycoproteomics experiments[80]. Thus, although linkage-specific and structure-specific information can be gleaned from glycopeptide MS/MS spectra[50,80,81], our analysis is restricted to the monosaccharide compositions reported by two widely used glycopeptide assignment tools (MSFragger-Glyco and Byonic). We want to emphasize, however, that OxoScan-MS data can be retrospectively mined for custom fragment ions of interest, including structure-specific oxonium ions. OxoScan-MS data can therefore be easily integrated with future developments in applying non-ubiquitous oxonium ions or fragment ion ratios for glycan classification, including those relating to clinically relevant glycan structures such as Lewis a/ Lewis x epitopes, rationally designed chemical probes or other endogenous post-translational modifications[82-87]. We finally note that caution should be exercised when inferring structure-specific information solely from oxonium ions, and further investigations (such as exoglycosidase treatments and structure-specific separations) are necessary for confirmation[88].

We anticipate that large-scale clinical glycoproteomic profiling, supported by increasingly high-throughput and quantitative glycoproteomics technologies, can aid in the discovery of glycoform-specific biomarkers relevant for understanding disease mechanisms as well as for diagnosis and prognosis. No enrichment steps were used in this study, enabling a workflow for clinical applications where reproducibility is of utmost importance. Importantly, omitting enrichment allows for parallel analysis of protein-level and peptide-level changes, which when integrated with glycopeptide quantification can help disentangle the multiple potential mechanisms of glycan regulation. However, we emphasize that the dynamic range and depth might be further increased by removing highly abundant proteins or via glycopeptide enrichment strategies. In the case that specific subsets of the glycoproteome are of specific interest, enrichment can also be coupled with optimized OxoScan-MS methods, for example, focused on immunoglobulin quantification. We also note that in the current study, we identified predominantly N-glycopeptides, but future optimization for O-glycan-derived fragment ions and O-glycan enrichment strategies could improve the detection of O-glycosylated peptides. This is a common trade-off in plasma (glyco)proteomics experiments; however, for our purposes, we focused on increasing the practical throughput and reducing costs of glycoproteomics experiments, thus incorporating minimal extra handling steps. We further note that although different LC−MS platforms were used for glycopeptide quantification and identification as proof-of-concept, next-generation mass spectrometers that integrate both scanning quadrupole capability and multiple complementary fragmentation strategies amenable to glycopeptide analysis will notably streamline the reported approach. Beyond biomarker discovery in plasma, we anticipate that OxoScan-MS could have a number of immediate applications, for example, in the high-throughput glycoprofiling of biologics and of the workhorse cell lines used to produce them.

## Methods

### Materials
LC−MS grade reagents were purchased as follows: water (Thermo Fisher, 10505904), acetonitrile (ACN, Thermo Fisher, 10001334), methanol (MeOH, Thermo Fisher, 10767665), formic acid (FA, Pierce, 85178), trifluoroacetic acid (TFA, Sigma-Aldrich, 85183), DL-dithiothreitol (DTT, Sigma-Aldrich, 43815), iodoacetamide (IAA, Sigma-Aldrich, I1149), urea (Sigma-Aldrich, 1084870500) and ammonium bicarbonate (ABC, Thermo Fisher, 15645440). Trypsin was purchased from Promega (V5117). Solid-phase extraction plates were purchased from NEST (BioPureSPN Macro 96-well, 100 mg PROTO 300 C18, HNS S18V-L).

### IgG isolation from human serum
IgG was purified from human serum samples as described previously[62]. In brief, IgG was isolated from 5 μl of serum using 30 μl of Protein A Sepharose (GE Healthcare). Sample mixtures were incubated under agitation at 650 r.p.m. for 1 h at room temperature. Protein A Sepharose beads were washed with 5 × 200 μl 1 × PBS and 3 × 200 μl MilliQ water. IgG was eluted with 3 × 100 μl 100 mM FA. Eluates were dried in a vacuum centrifuge, then redissolved in 50 μl 50 mM ammonium bicarbonate and shaken for 5 min. Sequencing-grade trypsin (Promega) was added to a final concentration of 0.2 μg μl$^{-1}$ and samples were incubated overnight at 37 °C. On the following day, IgG glycopeptides were isolated from peptides using self-made micro-spin cotton-HILIC columns. They were conditioned by washing with 3 × 50 μl MilliQ water and 3 × 50 μl 80% ACN. Afterwards, dried IgG samples were resuspended in 50 μl 80% ACN and loaded on the self-made microcolumns. They were washed with 3 × 50 μl 80% ACN containing 0.1% TFA and then with 3 × 50 μl 80% ACN. The retained IgG glycopeptides were eluted with 6 × 50 μl MilliQ water, dried out in a vacuum centrifuge and stored at −20 °C until measurement.

### Standard preparation of IgG and serum samples
Purified IgG (20 μg) or 5 μl of raw plasma/serum were prepared as previously described[5]. In brief, IgG/plasma was denatured and reduced by addition of 55 μl 8 M urea, 5.5 mM DTT and 100 mM ABC, followed by incubation for 1 h at 30 °C. All subsequent steps were carried out using a Beckman Coulter Biomek NXP 96-well liquid handling robot. IAA (5 μl 100 mM) was added and the mixture incubated in the dark for 30 min. Reduced/alkylated proteins were then diluted with 340 μl 100 mM ammonium bicarbonate (to bring [urea] to < 2 M) and digested with trypsin (1:50 w/w) for 17 h at 37 °C. Digestion was stopped by acidification with 25 μl 10% FA and peptides were cleaned up by solid-phase extraction (SPE) (NEST C18 MacroSPIN SPE plates, as described previously[21]). In brief, each well was treated/centrifuged sequentially in the following steps: 200 μl MeOH, 1 min at 50 g, 2 × 200 μl 50% ACN, 1 min at 150 g, 2 × 200 μl 0.1% FA, 1 min at 150 g, 200 μl sample, 1 min at 150 g, 2 × 200 μl 0.1% FA, 1 min at 200 g, 1 min at 200 g, 3 × 10 μl 50% ACN and 1 min at 200 g. Elution (50% ACN) fractions were eluted into the same respective wells and dried in an Eppendorf Speedvac (45 °C, ~7 h). Dried desalted peptides were resuspended in 0.1% FA (0.5–2 μg μl$^{-1}$, depending on sample) and stored at −80 °C until measurement.

### Glycosidase treatment
Deglycosylation was performed with the Protein Deglycosylation Mix II (New England Biosciences, P6044S). For glycosidase treatment, plasma samples were prepared as described above with the following modifications: following dilution of reduced/alkylated plasma with 340 μl 100 mM ABC, 45 μl 10X Protein Deglycosylation buffer I was added. Next, 5 μl of either Protein Deglycosylation Mix II (New England Biosciences, P6044S) or 100 mM ABC (for deglycosylation and control, respectively) were added and incubated at room temperature for 30 min and at 37 °C for a further 16 h. Following deglycosylation, tryptic digest and SPE was performed as described above. Dried samples were redissolved in 50 μl 0.1% FA and injected as is. Samples were measured with a 45 min water-to-acetonitrile gradient with a 10 m/z Scanning SWATH window (see Supplementary Table 4).

### Heavy-labelled E. coli growth and sample preparation
E. coli MG1665 was plated on LB agar and grown in M9 minimal media supplemented with $^{13}$C-glucose (11.28 g l$^{-1}$ M9 salts, 2 mM MgSO$_4$, 0.1 mM CaCl$_2$, 1% $^{13}$C-glucose). Cells were collected at mid-log phase, washed with water and lysed in 200 μl 7 M urea and 100 mM ABC with acid-washed glass beads (425–600 μm). Samples were then prepared as described previously[21]. Briefly, cells were lysed with mechanical bead beating (1600 MiniG, Spex Sample Prep) for 5 min at 1,500 r.p.m., reduced with 20 μl 55 mM DTT for 60 min at 30 °C and subsequently

alkylated with 20 µl 120 mM IAA at room temperature in the dark for 30 min. Lysates were then diluted with 1 ml 100 mM ABC, centrifuged at 3,220 g for 5 min and the supernatant taken for tryptic digest (9 µl 0.1 µg µl$^{-1}$ solution) for 17 h at 37 °C. Acidification and SPE clean-up was performed as described for plasma, with the following modifications: 3% ACN and 0.1% FA were used instead of 0.1% FA and elution volumes were 120 µl, 120 µl and 130 µl. Eluted peptides were dried and redissolved as described for plasma.

## Spike-in sample preparation

Commercial serum tryptic digests (prepared as described above) and heavy-labelled *E. coli* tryptic digests were resuspended in 0.1% FA and the peptide concentration measured on a Lunatic spectrophotometer. The digests were subsequently mixed in set ratios by protein amount (serum:*E. coli*; 5:95, 20:80, 40:60, 80:20), normalized to the same sample volume and 2 µg injected for each sample. Wiff files were then converted to .dia files in DIA-NN, extracted ion chromatograms (XICs) extracted (as .txt files) across the entire precursor range using the −extract [oxonium ion masses] function and the resulting output text files were directly imported into OxoScan scripts (as a Jupyter Notebook). The following settings were used for the spike-in method: maximum number of glycopeptide features called is 5,000, m/z bin width = 2 (m/z), retention time (RT) bin width = 0.025 min, m/z quantification radius = 5 (bins), RT quantification radius = 3 (bins), m/z exclusion radius = 2 × m/z quantification radius and RT exclusion radius = 3 × RT quantification radius.

## COVID-19 patient samples

Patient samples were obtained as part of the Pa-COVID-19 study, as described in detail previously[21,89]. Cohort demographics are shown in Supplementary Table 2. Thirty COVID-19 patients and 15 healthy controls were included in the COVID-19 study. Age of participants ranged from 22–86 (median 48) and patients were grouped into the following severity ratings using the WHO ordinal scale as follows: healthy, WHO 0, n = 15; mild, WHO 3, n = 10; moderate, WHO 4–5, n = 7; severe, WHO 6–7, n = 10. The Pa-COVID-19 study complies with the 1964 Declaration of Helsinki and later amendments. The study was approved by the Charité Ethics Committee (EA2/066/20) and where applicable was carried out in accordance with the principles of Good Clinical Practice (International Council for Harmonization, ICH 1996).

## COVID-19 cohort analysis

Patient samples were prepared as described in the general workflow and processed without further enrichment/depletion. The 45 biological samples were randomized into 96-well plate format and prepared in whole-process triplicate alongside aliquots of commercial plasma citrate. To minimize the effect of instrument drift, samples were block randomized by replicate for sample acquisition. A pooled plasma sample was generated by mixing a small aliquot of tryptic peptides from each clinical sample (mass spec QC, n = 10) and measured every 16 samples throughout the batch to monitor instrument performance. Commercial plasma was added to 96-well plates and prepared in parallel with the clinical samples as whole-process QCs (sample prep QC, n = 9). Blanks and mass calibration samples ('Pepcal') were also included every 16 injections across the cohort.

## Data-independent acquisition (OxoScan-MS)

All Scanning SWATH/DIA analysis was performed on a Waters NanoAcquity HPLC coupled to a Sciex TripleTOF 6600 mass spectrometer. Peptides were separated on a reverse-phase C18 Waters HSS T3 column (1.8 µm, 300 µm × 150 mm, 35 °C column temperature) at 5 µl min$^{-1}$ (loading flow/buffers). Peptides were separated with gradients of buffer A (1% ACN, 0.1% FA) and buffer B (ACN, 0.1% FA). The Cohort method ramped with a nonlinear gradient from 3–40% B over 19 min (Supplementary Table 3), while chromatographic gradients for glycosidase

treatment and gas-phase fractionation ramped linearly from 3–40% over 45 and 90 min, respectively. For IgG analysis, a linear gradient ramped from 3–18% buffer B over 90 min. Upon reaching 40% in the respective gradients, washing and re-equilibration steps were as follows: 40–80% B over 1 min, 80% B for 0.5 min, 80–3% B over 1 min, re-equilibration at 3% B for 6 min until next injection. Source conditions were as follows: source gas 1: 15 psi, source gas 2: 20 psi, curtain gas: 25 psi, temperature: 0 °C, IonSpray floating voltage: 5,500 V, declustering potential: 80 V. Rolling collision energies were calculated from the following equation: CE = 0.034 × m/z + 2, where m/z is the centre of the scanning quadrupole bin. Precursor range, window width and cycle times were tailored depending on chromatographic gradient, desired Q1 resolution and sensitivity (Supplementary Table 4).

## Data-dependent acquisition

Samples were pooled from all healthy and severely ill patients and analysed on an Orbitrap Eclipse mass spectrometer coupled to an Ultimate 3000 RSLCnano HPLC (both Thermo Fisher). Sample (1 µl, -1 µg µl$^{-1}$ in 0.1% FA) was loaded onto a trap column (Acclaim PepMap-100 75 µm × 2 cm NanoViper) with loading buffer (2% ACN, 0.05% TFA) at 7 µl min$^{-1}$ for 6 min (40 °C). Peptides were separated on an analytical column (PepMap RSLC C18, 75 µm × 50 cm, 2 µm particle size, 100 Å pore size, reversed-phase EASY-Spray, Thermo Fisher) from 2–40% buffer B over 87 min at 275 nl min$^{-1}$. The following parameters were used: column temperature: 40 °C, spray voltage: 2,400 V. Gradient elution buffers were: A: 0.1% FA, 5% DMSO and B: 0.1% FA, 5% dimethylsulfoxide (DMSO), 75% ACN. For MS scans acquired in the Orbitrap, scan resolution was set to 120,000 at FWHM (full width at half-maximum peak height) of 200 m/z. The precursor range was 400–2,000 m/z with the following parameters: RF lens 30%, AGC target 100%, maximum injection time 50 ms, spectra acquired in profile. Monoisotopic peak determination was set to the peptide mode. Dynamic exclusion was enabled to exclude previously selected precursor ions for 10 s after n = 3 times within 10 s, with mass tolerance of ±10 ppm. Precursors (z = 2–6) were selected for DDA MS/MS with a quadrupole isolation window of width 2 m/z and a fixed cycle time of 3 s. HCD MS/MS scans were acquired in the Orbitrap at a resolution of 30,000 and a normalized collision energy of 28% with the following parameters: first mass m/z 100, AGC target 100%, custom maximum injection time 54 ms, scan data acquired in centroid mode. An HCD-pd-ETD instrument method, whereby ETD fragmentation was only performed if three of the following list of mass trigger ions were present in the HCD MS/MS spectra (±20 ppm) and above the relative intensity threshold of 5% (126.055, 138.0549, 144.0655, 168.0654, 186.076, 204.0855, 366.1395, 292.1027, 274.0921, 657.2349 m/z). Precursor priority was given by highest charge state and ETD activation used calibrated charge-dependent ETD parameters. The single scan per cycle was detected in the ion trap with the following parameters: isolation window of 3 m/z, rapid scan rate, first mass m/z 100, AGC target 100%, custom maximum injection time 54 ms, scan data acquired in centroid mode.

## MRM-HR acquisition

Targeted mass-spectrometric analysis was conducted on a ZenoTOF 7600 mass spectrometer (AB Sciex) connected to a Waters Acquity M-class UPLC. The column setup and operating conditions were identical to the ones previously described (see 'Data-independent acquisition'), as were the MS settings with the following exceptions: buffer A was 0.1% FA, TOF-MS accumulation time of 0.25 s, TOF-MS scanning from 200–1,500 m/z at 10 eV CE, TOF-MS/MS using Zeno-pulsing with a threshold of 2 × 10$^5$ cps, then scanning from 100–1,500 m/z. Twenty-four glycopeptides, 30 unmodified peptides from the same protein, as well as 10 unrelated peptides for quality control were selected for MRM-HR following validation in preliminary analyses (details in Supplementary Table 6) based on overall retention time, expected fragment m/z (from DDA) and correlation thereof in several

iterations using an MRM-HR approach with relaxed retention time restraints and processing in Skyline 22.2 (glycopeptides)[90], or via comparison to SWATH acquisitions processed in DIA-NN (non-glycosylated precursors). Target-specific retention times for this LC–MS setup were corrected if necessary and defined with ±75 s tolerance in the final MRM-HR method. Target-specific collision energies were derived from the formula above (see 'Data-independent acquisition').

### DIA data processing

Raw Scanning SWATH data files (.raw) were processed to Sciex .wiff format using the Scanning SWATH raw processor (AB Sciex) with default settings except for the following: Q1 binning = 4. Wiff files were then converted to .dia files in DIA-NN and XICs were extracted (as .txt files) across the entire precursor range using the –extract [oxonium ion masses] function. The output text files were directly imported into OxoScan scripts (as a Jupyter Notebook). For the COVID-19 cohort method, the following settings were used: maximum number of glycopeptide features called is 5,000, $m/z$ bin width = 2 ($m/z$), RT bin width = 0.025 min, $m/z$ quantification radius = 5 (bins), RT quantification radius = 3 (bins), $m/z$ exclusion radius = 2 × $m/z$ quantification radius and RT exclusion radius = 3 × RT quantification radius. Samples were normalized and scaled before retention time alignment to prevent distortions due to variable sample loadings.

### Data analysis

All processed data (OxoScan/Byonic/MSFragger/Skyline output, exported MS data) were analysed using custom R scripts. General data manipulation was carried out with tidyverse packages[91] and visualization with ggplot2[92]. Differential expression analysis was performed with the limma R package[93] for generating paired comparisons between healthy and each disease grade, as in Extended Data Fig. 3d. The Kendall–Tau test was performed across WHO disease grades with the Theil–Sen trend estimator (as part of the EnvStats package[94]), followed by correction for multiple testing (Benjamini–Hochberg method) for significance analysis of specific glycopeptide changes with disease severity, as in Fig. 5, and Extended Data Figs. 5 and 6c. Sample sizes for each disease grade are described in Supplementary Table 2. Heat maps were plotted with the ComplexHeatmap R package[95]. PeakView (AB Sciex) was used for accessing raw MS data for precursor mass assignment, manual inspection and exporting of spectra/XICs.

All analysis scripts and figure generation can be reproduced at https://github.com/ehwmatt/OxoScan-MS. In brief, for each patient, a mean sample intensity and c.v. were calculated for each glycopeptide feature from three technical replicates and used for further analysis/statistical testing. Five samples were removed from the analysis due to low signal intensity and all samples were median normalized. To prevent misidentification of non-glycosylated precursors due to interfering signals in the oxonium ion regions, glycopeptide features for which a single oxonium ion comprised >85% of the total oxonium ion signal were removed. Furthermore, specific ion signals were removed if the percentage contribution for a given glycopeptide feature showed significant variability (indicating interference/poor quantitation). Finally, glycopeptide features were kept for quantification only if >3 oxonium ions were quantified across all samples in the clinical cohort. After these filtering steps, 1,002 glycopeptide features were kept for quantification.

### DDA data processing

Data-dependent glycoproteomics experiments were analysed in Byonic (Protein Metrics, v.4.1.5) and MSFragger-Glyco (v.3.7)[72,73].

For Byonic, .raw files were searched against the Uniprot Human FASTA (3AUP000005640-canonical, downloaded 26 May 2018) and a built-in library of 57 human plasma glycans, 132 human N-glycans and 9 human O-glycans, all set as 'rare1'. Carbamidomethylation (+57.0214) was set as a fixed modification and oxidation (+15.9949) as 'common1'.

Tryptic digest was selected (RK, 'C-terminal cutter', fully-specific, max. 1 missed cleavage). The following search parameters were applied: precursor tolerance: 5 ppm, fragment tolerance (HCD): 5 ppm, fragment tolerance (ETD): 0.6 Da, protein false-discovery rate (FDR): 1%. Identified glycopeptide information ('Spectra' tab of each Byonic output file) was imported into R and PSMs were further filtered with the following thresholds: presence of glycan in 'Glycans NHFAGNa' column, Byonic score > 150, |log Prob| > 3 (refs. 48,96).

For MSFragger, the default N-glycan and O-glycan hybrid search settings were loaded in Fragpipe 18.0 and used without modification (except in the case of semi-tryptic search for IGHA1 glycopeptides, commonly reported in the literature with a truncated C-terminal form[63] and also found in our Byonic data). Only identifications with a glycan $q$-value < 0.01 were kept.

The resulting identification table was taken forward for matching to identified DIA glycopeptide features with custom R scripts and manual validation, as described below.

### DIA high-resolution MS1 assignment

Prioritized glycopeptide features from the 167 putative matches between OxoScan-MS glycopeptide features and validated DDA assignments were selected initially from high-abundance features as proof-of-principle and subsequently expanded to encompass different glycoforms of already identified glycoproteins and highly differentially abundant glycopeptide features in the COVID-19 cohort. For this subset of 22 prioritized glycopeptide features, precursors were identified in pooled plasma samples using two MS methods (with the same chromatographic gradient and precursor range as the cohort):

1. Q1 method: 2 $m/z$ Scanning SWATH window and total cycle time of 3.6 s
2. MS1 method: MS1 scans only with 500 ms accumulation time

Precursor masses were identified by extracting oxonium ion chromatograms and Q1 profiles over the RT/binned precursor $m/z$ for specific glycopeptide features (either from a specific 'peak_num' in Supplementary Table 5 or a specific glycopeptide identified in DDA experiments) in the Q1 method. For each glycopeptide feature, the reported MS/MS spectra were exported directly for DDA/DIA comparison and fragment assignment. The respective accurate precursor $m/z$ was then extracted in the MS1 method with a tolerance of 0.1 Da and retention times matched to within 0.5 min. The MS1 spectra were exported directly from PeakView (AB Sciex). High-resolution precursor $m/z$ values were used to calculate precursor mass and matched to Byonic-reported glycopeptide precursors with a tolerance of 0.5 Da. Q1 profiles were further inspected for each glycopeptide feature analysed with a narrow-window (2 $m/z$) OxoScan-MS method and any features with nearby (5 $m/z$) co-eluting glycopeptides were removed.

### MS/MS matching and glycopeptide validation

To compare DDA and DIA MS/MS spectra, both HCD spectra and fragment ion assignments from each identified glycopeptide were exported from Byonic as text files. Extracted Scanning SWATH MS and MS/MS spectra (as described above) were exported as text files. Matching fragments were compared between DDA/DIA spectra with a custom R script. For MS/MS matching between DDA/DIA experiments, a list of theoretical and observed fragment ions was exported directly from Byonic for each glycopeptide feature. DDA spectra were matched first to the Byonic fragment list with a tolerance of 20 ppm and subsequently with the DIA MS/MS spectra with a tolerance of 20 ppm. In the case of multiple matches, only the match with the lowest mass error was taken.

### Normalization of MRM-HR measurements

No batch or sample normalization was applied to individual glycopeptide/peptide measurements; instead, all glycopeptide abundances were scaled to their respective adjacent/unmodified peptides. For

adjacent peptides (those from the same protein group as their respective glycopeptides), two or more unmodified peptides were quantified in the MRM-HR method. Glycopeptide abundances were then normalized to either the mean peptide intensities (for adjacent peptides) or single peptide intensities (for unmodified peptides) from the same samples.

## Reporting summary

Further information on research design is available in the Nature Portfolio Reporting Summary linked to this article.

## Data availability

Raw MS data (OxoScan-MS, DDA and MRM-HR), extracted oxonium ion. txt files from DIA-NN and OxoScan-MS processed outputs are available via MassIVE on ProteomeXchange (accession number: PXD034172). OxoScan-MS (Scanning SWATH) data can be opened in PeakView (AB Sciex) with a suitable license and via Skyline. Source data for the figures in this study are available in figshare with the identifier https://doi. org/10.6084/m9.figshare.c.6677135.v1 (refs. 97,98). All processed data and accompanying scripts are also available on Zenodo at https://doi. org/10.5281/zenodo.8015483.

## Code availability

All custom code (OxoScan Python functions/Jupyter notebooks and R scripts for analysis and for reproducing all figures) and OxoScan-MS processed data for IgG, spike-in experiment and the COVID-19 cohort are freely available at https://github.com/ehwmatt/OxoScan-MS. Code with all accompanying processed data is also available on Zenodo at https://doi.org/10.5281/zenodo.8015483.

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

## Acknowledgements

We thank L. Sander, M. Witzenrath and W. Kuebler (Charité Universitaetsmedizin Berlin), as well as all members of the PA-COVID-19 study group for joint work on the COVID-19 studies; the organizers and all collaborators at the 2020 Crick Data Challenge, which stimulated the strategy of oxonium ion quantification; S. Kamrad for providing *E. coli* samples for the plasma dilution experiment; and the Charité Core Facility High Throughout Mass Spectrometry, especially Daniela Ludwig, for support in sample and data generation. Figures 2a and 3a were created with BioRender.com.

## Author contributions

M.E.H.W., C.B.M. and M.R. designed the study. M.E.H.W. and L.K. prepared samples for glycoproteomic analysis. M.E.H.W., C.B.M., L.R.S. and H.R.F. carried out mass-spectrometry experiments. M.M., Z.W. and V.B. provided input on mass spectrometric method set-up and development. D.M.J., J.d.F., S.K.A., M.E.H.W. and C.B.M. developed the OxoScan Python analysis approach. M.E.H.W., C.B.M., V.D., D.M.J. and L.R.S. analysed the data. P.T.-L. and F.K. collected COVID-19 clinical samples. M.E.H.W., C.B.M., L.R.S. and M.R. wrote the paper, with input from all co-authors.

## Funding

 This work was supported by the Francis Crick Institute, which receives its core funding from Cancer Research UK (FC001134), the UK Medical Research Council (FC001134) and the Wellcome Trust (FC001134). Part of this research was funded by the European Research Council (ERC) under grant agreement ERC-SyG-2020 951475, the Wellcome Trust (IA 200829/Z/16/Z), and by the Ministry of Education and Research (BMBF), as part of the National Research Node 'Mass spectrometry in Systems Medicine (MSCoresys) under grant agreement 161L0221 & 031L0220. C.B.M. was supported by the Precision Proteomic Center Davos which receives funding through the Swiss canton of Grisons. L.K. was supported by the German Research Foundation.

## Competing interests

M.R. is founder and shareholder of Eliptica Ltd.

## Additional information

**Extended data** is available for this paper at https://doi.org/10.1038/s41551-023-01067-5.

# Article

**Correspondence and requests for materials** should be addressed to Christoph B. Messner or Markus Ralser.

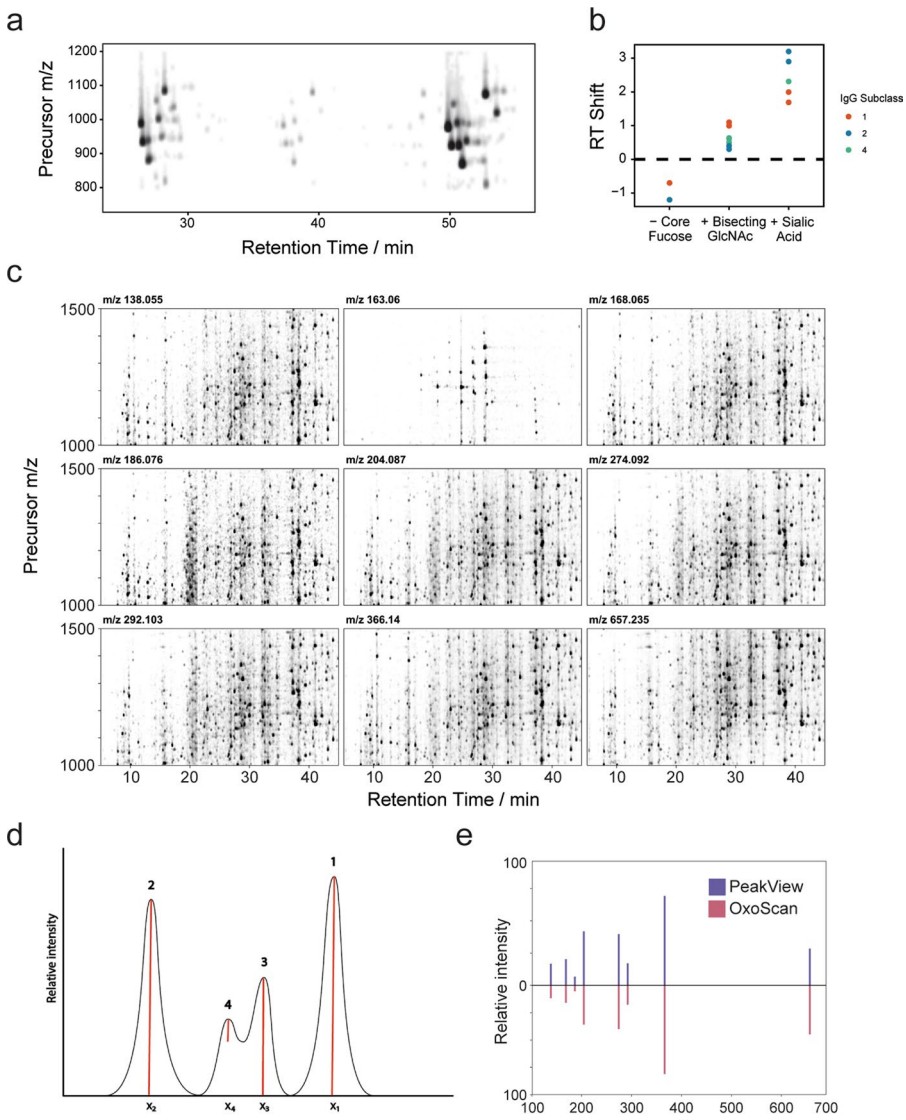

**Extended Data Fig. 1 | Qualitative and quantitative glycoproteomic analysis by OxoScan-MS. a.** Oxonium ion map of purified IgG from human serum[1], showing different total abundances of IgG 1, 4 and 2 subclasses, from left to right. Oxonium ion signals were extracted in DIA-NN[2], summed and plotted with opacity proportional to intensity. **b.** Retention time shifts in reverse-phase (C18) chromatography of identified IgG glycopeptides upon change of glycan composition, when compared to respective GXF (reference) glycopeptides. **c.** Oxonium ion maps of a human tryptic digest for 9 oxonium ions, extracted in DIA-NN (with a 20 ppm mass tolerance) and point opacity plotted proportional

to intensity (scaled separately by ion). **d.** Schematic showing the order of priority for peak calling (in 1-dimension) by the persistent homology algorithm. Peak numbering shows rank of persistence values and red lines represent the computed persistence value for each peak. Importantly, peaks are ranked by persistence as opposed to maximum height. **e.** Back-to-back MS/MS spectra of an IgG glycopeptide showing intensities of 8 oxonium ions when exported directly from the MS/MS spectrum (blue, top panel) in PeakView (AB Sciex) compared to output values from OxoScan quantification (red, bottom panel).

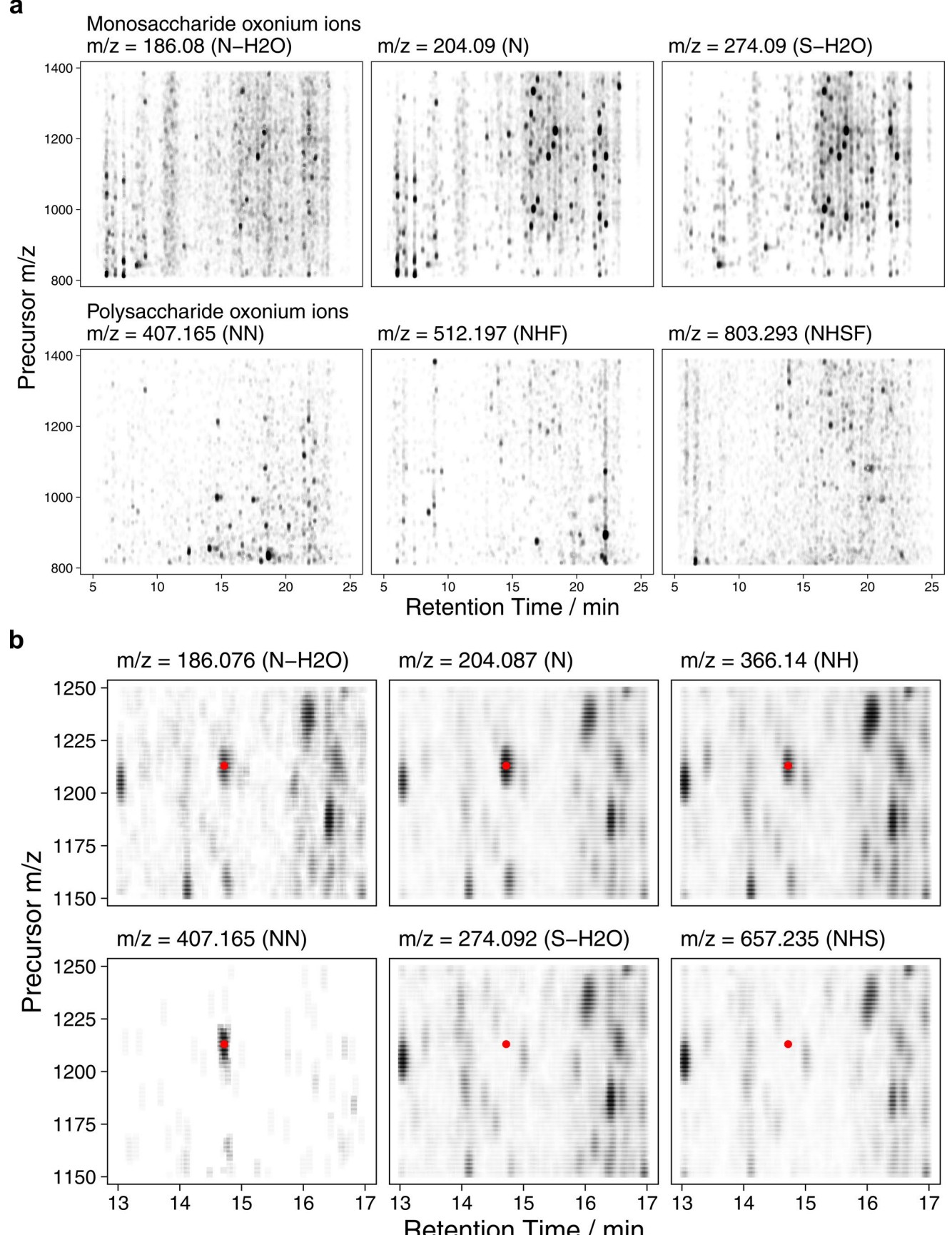

**Extended Data Fig. 2 | OxoScan-MS allows for retrospective extraction of custom ions of interest. a.** Figure shows full width 2-dimensional oxonium ion maps for a plasma sample measured in the COVID-19 cohort, with 3 common oxonium ions (*m/z* 186.076, 204.087, 274.092) from single monosaccharide units and more specific ions corresponding to 2-4 saccharide units (N = HexNAc, H = Hex, S = Neu5Ac, F = Fucose). **b.** Example glycopeptide feature showing co-localisation of HexNAc-HexNAc oxonium ion (*m/z* 407.165) with common oxonium ions, although notably Neu5Ac-derived oxonium ions are absent.

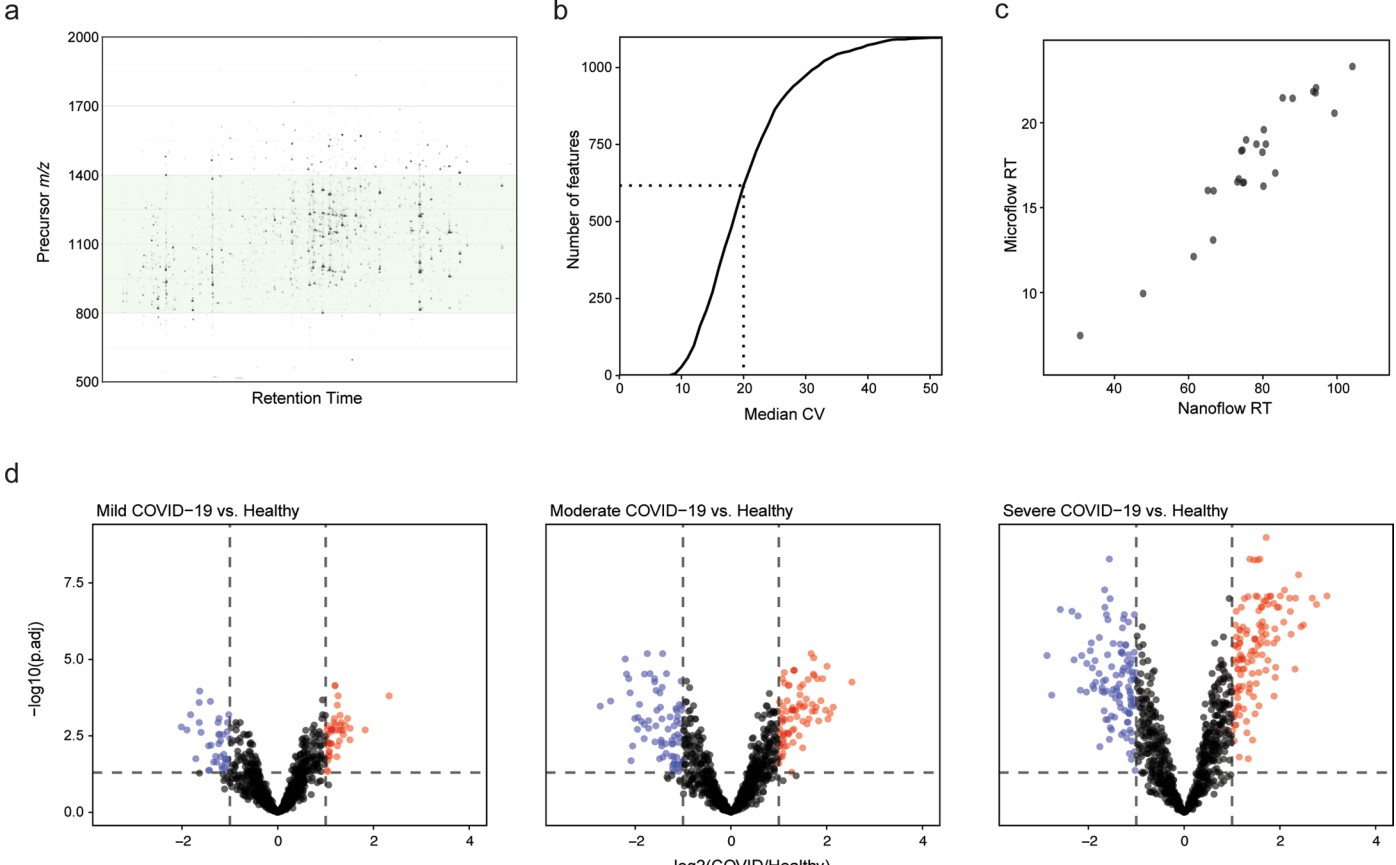

**Extended Data Fig. 3 | Profiling the glycoproteomic changes in SARS-CoV-2 infection by OxoScan-MS. a**. Gas-phase fractionation of a single commercial plasma tryptic digest over the precursor range $m/z$ 500-2000 (in 3 separate runs, shown aggregated here) shows the optimum range for detection of glycopeptides by OxoScan-MS. **b**. Median CV (%) values for each feature quantified in clinical samples. CVs were calculated for each feature in triplicate measurements of each patient/donor sample, the median taken for each feature, ranked and plotted against feature number. Dotted line shows the CV = 20% threshold. **c**. Comparison of retention times for glycopeptides identified in both DDA (nano-flow, x axis) and DIA (micro-flow, y-axis) shows good agreement across different chromatographic platforms. **d**. Volcano plots comparing $\log_2$(fold-change) for all glycopeptide features between each grouped disease severity (mild, moderate, severe) against healthy controls. $\log_2$(fold-change) and p-values were calculated using the limma R package[3]. Multiple testing correction was performed by the Benjamini-Hochberg method[4]. Coloured points represent those with $|\log_2$(fold-change)$| > 1$ and $P < 0.05$ for up- and down-regulated features (red and blue respectively).

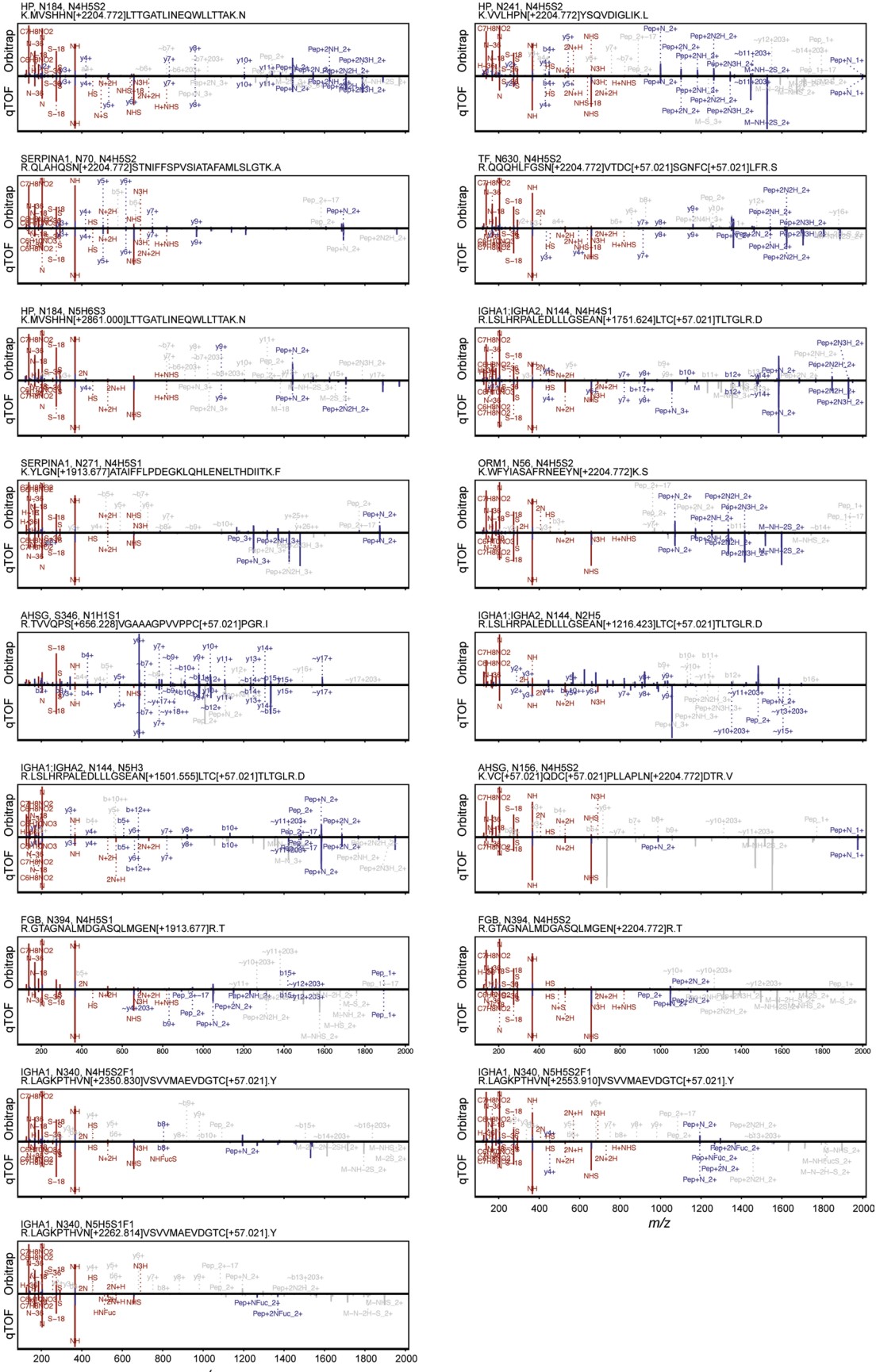

**Extended Data Fig. 4 | Comparison of MS/MS spectra from both Orbitrap and qTOF instruments.** Back-to-back comparison of DDA (top panels, HCD, Orbitrap, 1.6 m/z window) and DIA (bottom panels, CID, qTOF, 2 m/z window) MS/MS spectra for each of the candidate glycopeptides from the COVID-19 cohort.

For CID/HCD spectra, fragments matched to theoretical fragments exported from Byonic for each DDA spectrum are shown (0.1 Da tolerance). Fragments shared between DDA and DIA spectra are shown in blue, oxonium ions in red and singly-assigned fragments in grey.

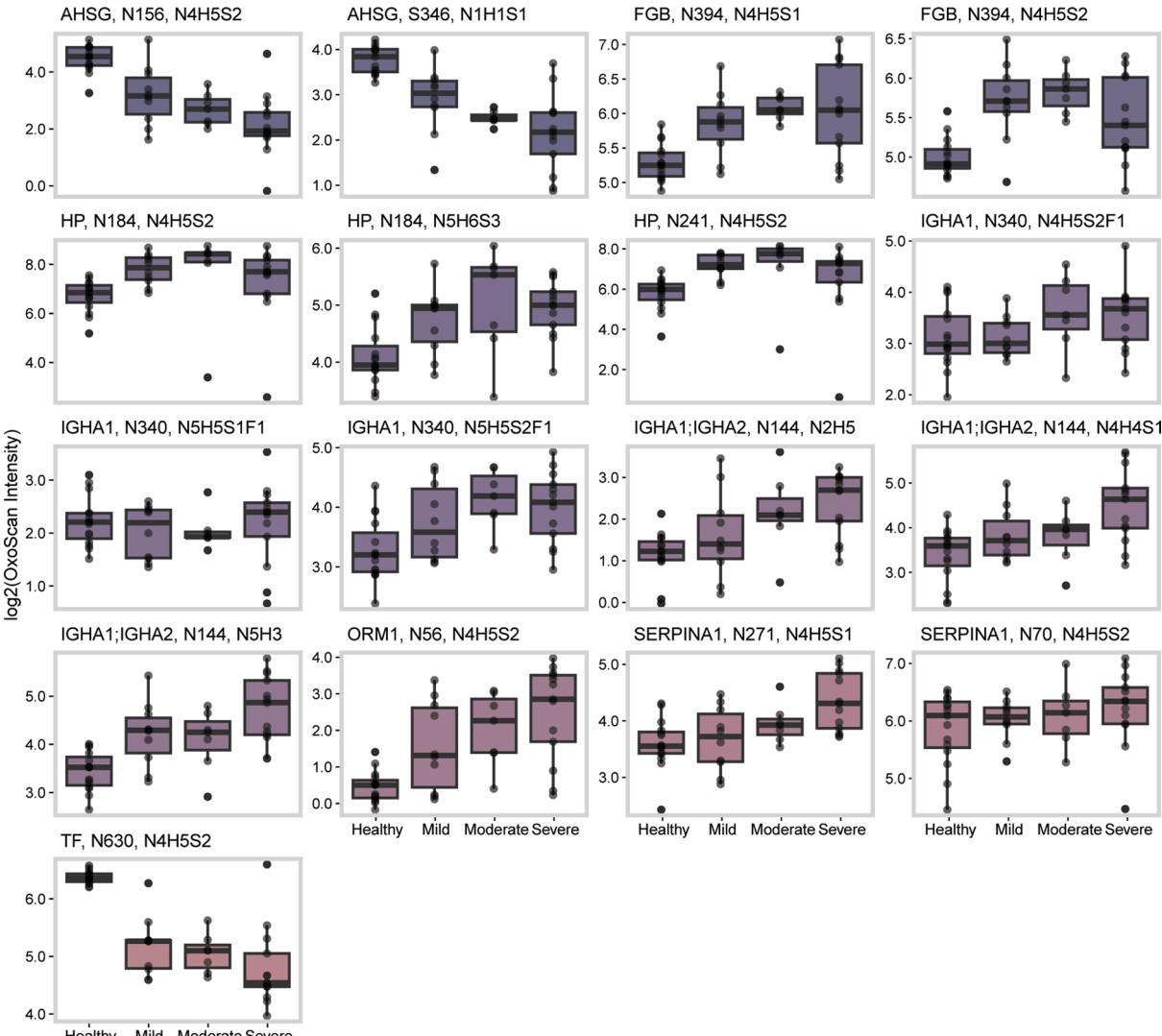

**Extended Data Fig. 5 | Severity-specific changes in glycopeptide feature abundance in COVID-19 patient plasma.** Abundances of glycopeptides identified in the COVID-19 cohort, grouped by disease severity. Values are log$_2$-transformed, box-and-whisker plot displays 25th, 50th (median) and 75th percentile in the box. Whiskers display upper/lower limits of data. Plot labels show gene, glycosylation site and glycan composition.

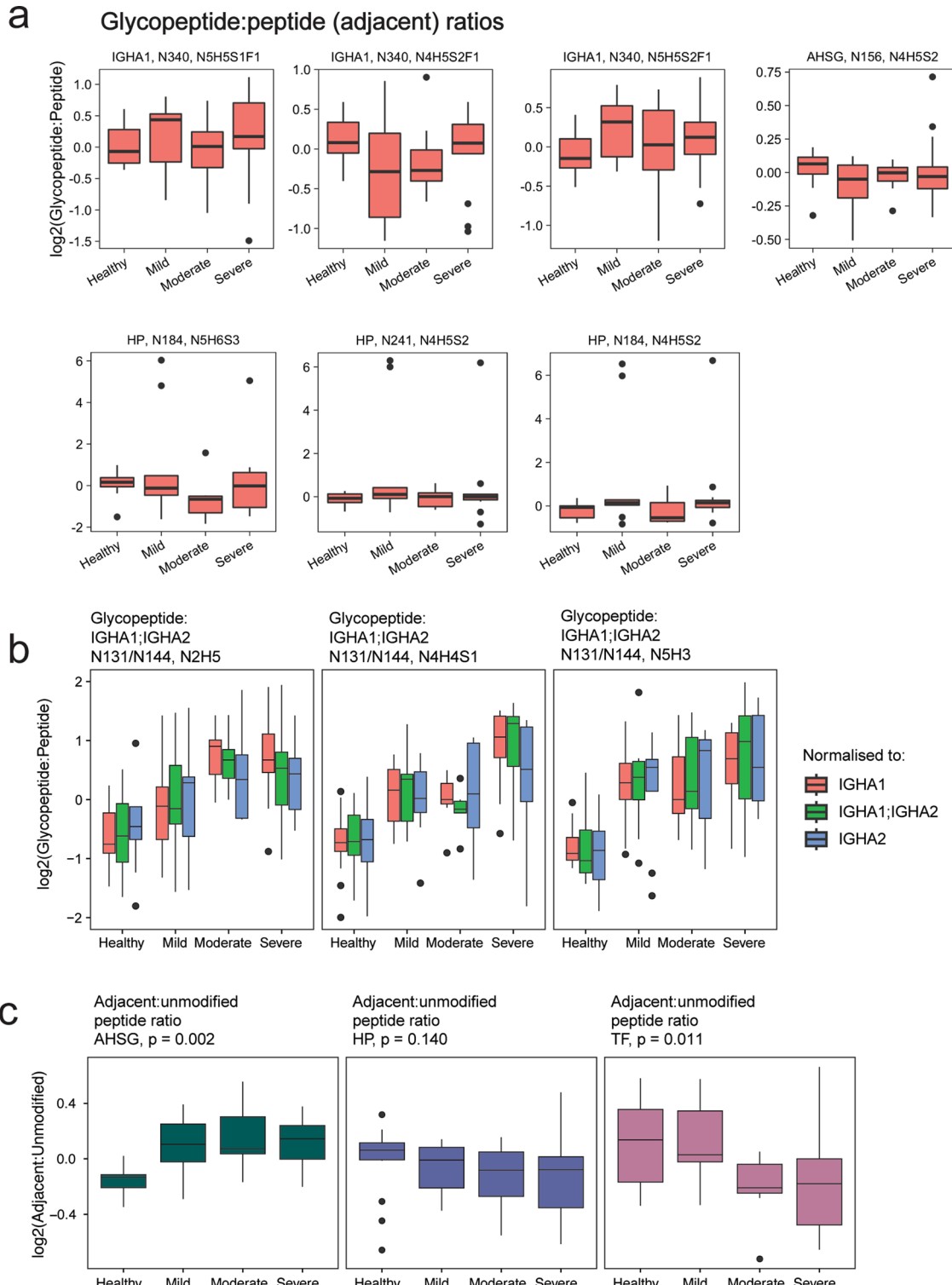

**Extended Data Fig. 6 | Normalization of glycopeptide abundances to peptide-level measurements. a**. Ratios of glycopeptide:adjacent peptides across COVID-19 severity classes, measured by parallel MRM-HR of both glycopeptide and adjacent peptides. **b**. Normalisation of IgA glycopeptides is robust to different subclasses (IGHA1, IGHA2). **c**. Non-modified peptides corresponding to measured glycopeptides (AHSG S346, TF N630) may vary differently to adjacent (containing no glycosite) peptides.

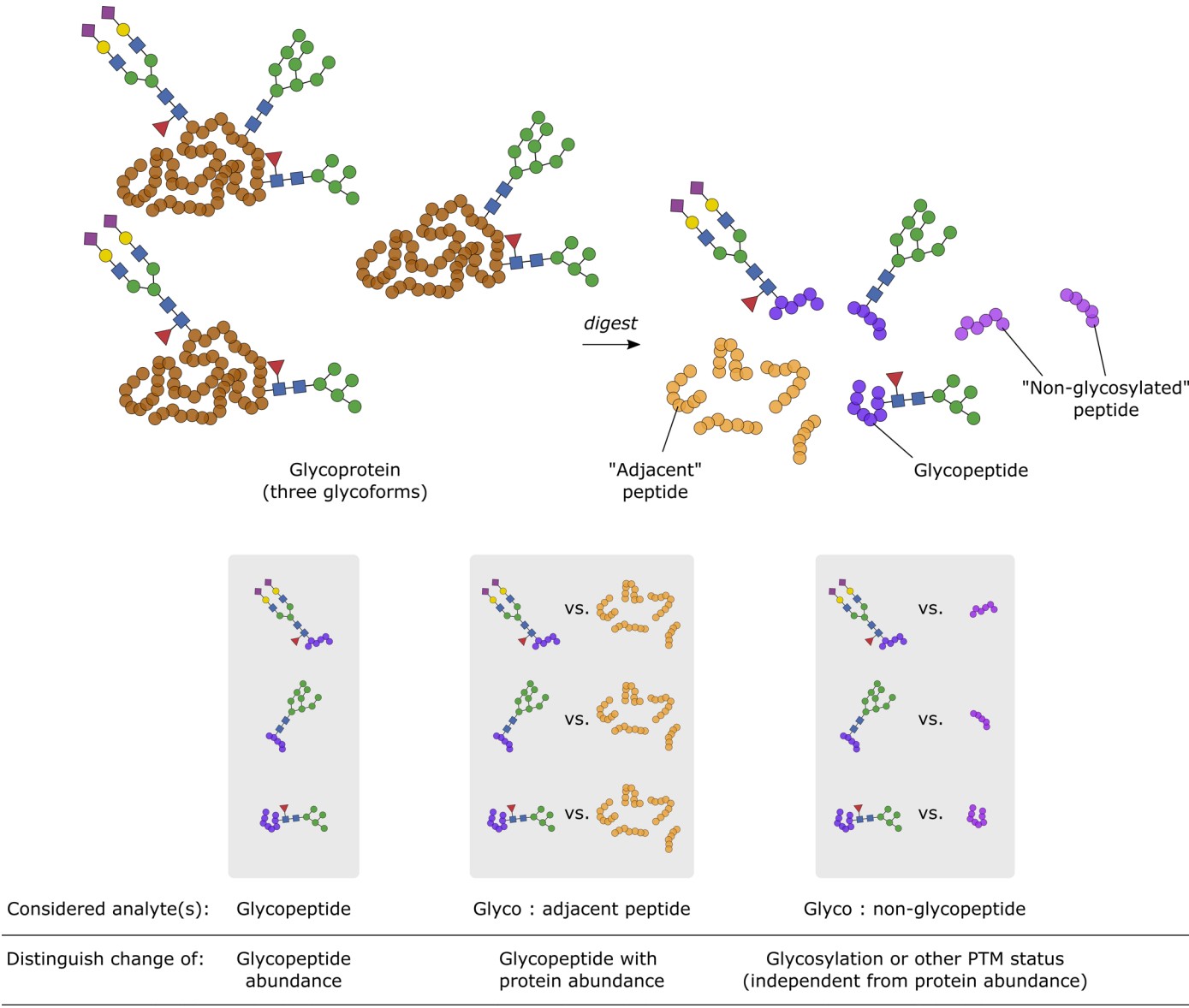

| Considered analyte(s): | Glycopeptide | Glyco : adjacent peptide | Glyco : non-glycopeptide |
|---|---|---|---|
| Distinguish change of: | Glycopeptide abundance | Glycopeptide with protein abundance | Glycosylation or other PTM status (independent from protein abundance) |

**Extended Data Fig. 7 | Schematic of glycan regulation inference.** Proteolysis of glycoproteins leads to glycosylated and unmodified peptides ("non-glycosylated" = unmodified peptidoform containing a glycosite, "adjacent" = unmodified peptide elsewhere within the protein sequence) that can be compared to distinguish between protein abundance and glycosylation status changes.

# Reporting Summary

## Statistics

For all statistical analyses, confirm that the following items are present in the figure legend, table legend, main text, or Methods section.

| n/a | Confirmed | |
|---|---|---|
| ☐ | ☒ | The exact sample size (*n*) for each experimental group/condition, given as a discrete number and unit of measurement |
| ☐ | ☒ | A statement on whether measurements were taken from distinct samples or whether the same sample was measured repeatedly |
| ☐ | ☒ | The statistical test(s) used AND whether they are one- or two-sided *Only common tests should be described solely by name; describe more complex techniques in the Methods section.* |
| ☒ | ☐ | A description of all covariates tested |
| ☐ | ☒ | A description of any assumptions or corrections, such as tests of normality and adjustment for multiple comparisons |
| ☐ | ☒ | A full description of the statistical parameters including central tendency (e.g. means) or other basic estimates (e.g. regression coefficient) AND variation (e.g. standard deviation) or associated estimates of uncertainty (e.g. confidence intervals) |
| ☐ | ☒ | For null hypothesis testing, the test statistic (e.g. *F*, *t*, *r*) with confidence intervals, effect sizes, degrees of freedom and *P* value noted *Give P values as exact values whenever suitable.* |
| ☒ | ☐ | For Bayesian analysis, information on the choice of priors and Markov chain Monte Carlo settings |
| ☒ | ☐ | For hierarchical and complex designs, identification of the appropriate level for tests and full reporting of outcomes |
| ☒ | ☐ | Estimates of effect sizes (e.g. Cohen's *d*, Pearson's *r*), indicating how they were calculated |

*Our web collection on statistics for biologists contains articles on many of the points above.*

## Software and code

Policy information about availability of computer code

| Data collection | Analyst 1.8.1 TF (AB Sciex), ScanningSWATH RAW Converter (AB Sciex), SCIEX OS 3.0.0.3363 & Waters ACQUITY Console Firmware 1.56 (MRM-HR data), Tune 3.4 & Xcalibur 4.4 & SII 1.6 for Xcalibur (Orbitrap DDA data). |
|---|---|
| Data analysis | DIA-NN 1.8, python 3.8.8, OxoScan-MS associated library (see https://github.com/ehwmatt/OxoScan-MS), R 4.2.2, RStudio 1.2.5019, tidyverse 1.3.2, ggplot2 3.4.0, limma 3.54.1, EnvStats 2.7.0, ComplexHeatmap 2.14.0, Byonic 4.1.5, MSFragger-Glyco 3.7, FragPipe 18.0. <br><br> All custom code (OxoScan Python functions/Jupyter notebooks and R scripts for analysis and for reproducing all figures) and OxoScan-MS processed data for IgG, spike-in experiment and the COVID-19 cohort are freely available at https://github.com/ehwmatt/OxoScan-MS. Code with all accompanying processed data is also available at Zenodo (DOI: https://doi.org/10.5281/zenodo.8015483). |

For manuscripts utilizing custom algorithms or software that are central to the research but not yet described in published literature, software must be made available to editors and reviewers. We strongly encourage code deposition in a community repository (e.g. GitHub). See the Nature Portfolio guidelines for submitting code & software for further information.

## Data

Policy information about availability of data

All manuscripts must include a data availability statement. This statement should provide the following information, where applicable:
- Accession codes, unique identifiers, or web links for publicly available datasets
- A description of any restrictions on data availability
- For clinical datasets or third party data, please ensure that the statement adheres to our policy

Raw MS data (OxoScan-MS, DDA and MRM-HR), extracted oxonium ion .txt files from DIA-NN and OxoScan-MS processed outputs are available via MassIVE on ProteomeXchange (accession number: PXD034172). OxoScan-MS (Scanning SWATH) data can be opened in PeakView (AB Sciex) with a suitable license, and via Skyline. Source data for the figures in this study are available in figshare with the identifier https://doi.org/10.6084/m9.figshare.c.6677135.v1 (ref. 98). All processed data and accompanying scripts are also available from Zenodo at https://doi.org/10.5281/zenodo.8015483.

## Research involving human participants, their data, or biological material

Policy information about studies with human participants or human data. See also policy information about sex, gender (identity/presentation), and sexual orientation and race, ethnicity and racism.

| | |
|---|---|
| Reporting on sex and gender | Information on biological sex was derived from self-reporting, and not considered in our analysis. |
| Reporting on race, ethnicity, or other socially relevant groupings | Information on race or ethnicity or any other socially relevant grouping was not collected and thus not considered in our analysis. |
| Population characteristics | 30 COVID-19 patients and 15 healthy individuals were included in the study. 47% (21/45) were female and 53% (24/45) were male. The median age was 50 (range 21–86). The severity was graded according to the WHO ordinal outcome scale of clinical improvement, with 15 with grade 0, 10 with grade 3, 4 with grade 4, 3 with grade 5, 3 with grade 6, and 10 with grade 7. |
| Recruitment | Sampling was performed as part of the Pa-COVID-19 study, a prospective observational cohort study assessing pathophysiology and clinical characteristics of patients with COVID-19 at Charité Universitätsmedizin Berlin (Kurth et al. Infection 2020). All patients with SARS-CoV-2 infection proven by positive PCR from respiratory specimens and willing to provide written informed consent were eligible for inclusion. Exclusion criteria were refusal to participate in the clinical study by the patient or legal representative, or clinical conditions that did not allow for blood sampling. The patients were hospitalized at Charité in Berlin between 1st and 26th of March 2020. |
| Ethics oversight | The Pa-COVID-19 study was carried out according to the Declaration of Helsinki, and the principles of Good Clinical Practice (ICH 1996) where applicable. The study was approved by the ethics committee of Charité-Universitätsmedizin Berlin (EA2/066/20). |

Note that full information on the approval of the study protocol must also be provided in the manuscript.

# Field-specific reporting

Please select the one below that is the best fit for your research. If you are not sure, read the appropriate sections before making your selection.

☒ Life sciences ☐ Behavioural & social sciences ☐ Ecological, evolutionary & environmental sciences

For a reference copy of the document with all sections, see nature.com/documents/nr-reporting-summary-flat.pdf

# Life sciences study design

All studies must disclose on these points even when the disclosure is negative.

| | |
|---|---|
| Sample size | Sample sizes were not predetermined on the basis of statistical methods.<br><br>For the proof-of-principle IgG experiments, a single replicate was used, as multiple comparisons (e.g. RT shifts for multiple IgG subclasses) could be observed in single samples.<br><br>Controlling the specificity of OxoScan with human plasma +/- deglycosylation used one replicate each, as the deglycosylation served as a strong negative control (that is, a very strong effect size).<br><br>Repeatability of glycopeptide quantitation with human plasma used 2 replicates.<br><br>Quantitative performance assessment via E. coli spike-in to human serum was measured as single replica per dilution.<br><br>Each sample of the COVID-19 cohort was produced and measured in OxoScan-MS mode in triplicate.<br><br>Glycopeptide ID from two pooled plasma samples (healthy and ill) by Orbitrap MS used one replica each. |

Validation by MRM-HR used single replicate injections of a different sample preparation starting from the same original patient plasma samples.

Data exclusions | No data were excluded.

Replication | Benchmarks were acquired in triplicates. COVID-19 cohort samples were prepared and measured as triplicates in OxoScan-MS mode, followed by validation by MRM-HR on another LC-MS platform (Waters M-Class + ZenoTOF 7600), in another lab. The findings were successfully reproduced.

Randomization | Samples were block-randomized whenever possible. The COVID-19 cohort samples were randomized.

Blinding | Measurements and analysis were not blinded, as there was no observer bias expected in this technical study.

# Reporting for specific materials, systems and methods

We require information from authors about some types of materials, experimental systems and methods used in many studies. Here, indicate whether each material, system or method listed is relevant to your study. If you are not sure if a list item applies to your research, read the appropriate section before selecting a response.

## Materials & experimental systems

| n/a | Involved in the study |
|---|---|
| ☒ | Antibodies |
| ☒ | Eukaryotic cell lines |
| ☒ | Palaeontology and archaeology |
| ☒ | Animals and other organisms |
| ☒ | Clinical data |
| ☒ | Dual use research of concern |
| ☒ | Plants |

## Methods

| n/a | Involved in the study |
|---|---|
| ☒ | ChIP-seq |
| ☒ | Flow cytometry |
| ☒ | MRI-based neuroimaging |

