## [Peer Review File · Nature Biomedical Engineering]

Oxonium-ion scanning mass spectrometry for large-scale plasma glycoproteomics

Corresponding author: Markus Ralser

Editorial note

This document includes relevant written communications between the manuscript's corresponding author and the editor and reviewers of the manuscript during peer review. It includes decision letters relaying any editorial points and peer-review reports, and the authors' replies to these (under 'Rebuttal' headings). The editorial decisions are signed by the manuscript's handling editor, yet the editorial team and ultimately the journal's Chief Editor share responsibility for all decisions.

Any relevant documents attached to the decision letters are referred to as **Appendix #**, and can be found appended to this document. Any information deemed confidential has been redacted or removed. Earlier versions of the manuscript are not published, yet the originally submitted version may be available as a preprint. Because of editorial edits and changes during peer review, the published title of the paper and the title mentioned in below correspondence may differ.

Correspondence

Sun 21 Aug 2022

Decision on Article nBME-22-1691-T

Dear Prof Ralser,

Thank you again for submitting to *Nature Biomedical Engineering* your manuscript, "OxoScan-MS: Oxonium ion scanning mass spectrometry facilitates plasma glycoproteomics in large scale". The manuscript has been seen by three experts, whose reports you will find at the end of this message. You will see that the reviewers appreciate the work, and that they raise a number of technical criticisms that we hope you will be able to address. In particular, and in addition to addressing the technical questions, we would expect that a revised version of the manuscript provides:

- * Clear description of the limitations of the technique.
- * Thorough methodological reporting, as per the various pertinent comments from the reviewers.

When you are ready to resubmit your manuscript, please upload the revised files, a point-by-point rebuttal to the comments from all reviewers, the reporting summary, and a cover letter that explains the main improvements included in the revision and responds to any points highlighted in this decision.

Please follow the following recommendations:

- * Clearly highlight any amendments to the text and figures to help the reviewers and editors find and understand the changes (yet keep in mind that excessive marking can hinder readability).
- * If you and your co-authors disagree with a criticism, provide the arguments to the reviewer (optionally, indicate the relevant points in the cover letter).
- * If a criticism or suggestion is not addressed, please indicate so in the rebuttal to the reviewer commentsand explain the reason(s).

* Consider including responses to any criticisms raised by more than one reviewer at the beginning of the rebuttal, in a section addressed to all reviewers.

* The rebuttal should include the reviewer comments in point-by-point format (please note that we provide all reviewers will the reports as they appear at the end of this message).

* Provide the rebuttal to the reviewer comments and the cover letter as separate files.

We hope that you will be able to resubmit the manuscript within 12 weeks from the receipt of this message. If this is the case, you will be protected against potential scooping. Otherwise, we will be happy to consider a revised manuscript as long as the significance of the work is not compromised by work published elsewhere or accepted for publication at *Nature Biomedical Engineering*.

We hope that you will find the referee reports helpful when revising the work, which we look forward to receive. Please do not hesitate to contact me should you have any questions.

Best wishes,

Pep

Pep Pàmies
Chief Editor, Nature Biomedical Engineering

Reviewer #1 (Report for the authors (Required)):

White et al. present a new data-independent glycoproteomics method - OxoScan-MS. This methodology is another application in the growing field of data-independent glycoproteomics. OxoScan was developed on the analysis of isolated IgG glycopeptides from human plasma. The methodology shows promising performance, but oxonium ions are the glycopeptide fragments with the lowest selectivity, so the performance of the methodology for low-represented glycopeptides in high-abundance background remains questionable and needs further testing. The authors tested the methodology on plasma samples from a cohort hospitalized for COVID-19. The authors claim to have obtained quantitative results for more than 1k glycopeptides. For the above reason, I recommend verifying the results of significantly changed glycopeptides with a more specific PRM methodology. Over all, this is a valid piece of glycoproteomics work and I recommend that this manuscript be published in a Nature Biomedical Engineering journal after major revision. I just have a few questions:

- a. How did the authors work with less abundant but more structured informative oxonium ions such as 407 (LacdiNAc etc.)? Do the authors plan to extend the methodology for site- and structure-specific data-independent glycoproteomics?
- b. The authors analyzed protein abundance in addition to glycopeptide abundance to normalize the data. How do the authors address glycosite occupancy, could this information be provided in Figure 5?
- c. In Figure 5, panel c shows the IgA1 and IgA2 shared glycopeptide changes, but which form of IgA are the protein changes

Reviewer #2 (Report for the authors (Required)):

This well written manuscript describes the development, validation and application of a new strategy, OxoScan-MS, for unbiased large-scale identification and relative quantification of tryptic glycopeptides,

obtained without enrichment steps, from glycoproteins of biological fluids e.g. plasma or serum, within a short time frame suitable for patient cohort studies performed either at university core or clinical reference laboratories.

The strategy is built on the already well-known characteristic positive – oxonium - ion fragmentation of glycans obtained from MS/MS fragmentation through CID and HCD, but it is here taken to a systematic screening level with DIA-SWATH-MS using fast LC C18 NanoAcquity chromatography and a Sciex triple TOF 6600 instrument and a newly developed, now freely available software OxoScan, to create 2D Oxonium ion profiles containing thousands of identifiable glycopeptides from a single sample. For complete structural identification of glycopeptides pooled samples were analyzed on another nLC-MS/MS instrument (Ultimate 300 RSLCnano linked to an Orbitrap Eclipse run in HCD-pd-ETD) and data were interpreted using the Byonic software.

After validation of the method on standard protein mixtures and plasma samples the OxoScan-MS was applied to analyze glycopeptides of 30 plasma samples of patients with mild to severe COVID-19 versus 15 healthy controls. Among 1.102 unique glycopeptide features in all samples, 90 % (1.002) were quantifiable across all clinical samples and spanned over four orders of magnitude. Data were bioinformatically digested to heatmaps for each individual and each glycan feature as well as with Principal Component analysis and also compared with proteomic analyses of the same samples giving an appreciation of which of 26 selected glycoproteins that significantly changed their concentrations (e.g. alpha-2-HS-glycoprotein and hemopexin) or which changed their site-specific glycosylation profiles (e.g. IgA, alpha-1-antitrypsin as well as fibrinogen alpha and beta chains) related to the (stage of) COVID-19 disease. Although complex patterns and individual variabilities were obtained, clear tendencies separating healthy from diseased individuals were revealed and are worthy of further studies.

The OxoScan-MS method shows great promise and will most certainly be welcomed as a useful tool in the search for complex patterns of disease biomarkers carried on glycoproteins of body fluids and of cultured cell and tissue media.

I have only a few mainly technical comments for the authors.

1. I think it is important to point out, as a limitation of the study, that the structural data provided are giving monosaccharide compositions of glycans rather than glycan structures. The authors use the term “features” – which as some early point in the text should be explained in structural terms.
2. In this respect is also important to discuss and reference in the text (preferentially in the Discussion) the weakness of the Byonic software in interpreting isobaric m/z peaks into composition or even structures (See ref Kawahara R, et al Community evaluation of glycoproteomics informatics solutions reveals high-performance search strategies for serum glycopeptide analysis. *Nat Methods*. 2021 Nov;18(11):1304-1316. doi: 10.1038/s41592-021-01309-x. Epub 2021 Nov 1. Erratum in: *Nat Methods*. 2021 Dec 10; Erratum in: *Nat Methods*. 2022 Jan;28(1):214. PMID: 34725484; PMCID: PMC8566223.). For solving this issue either manual interpretation with well defined criteria or alternative software should be used for structural confirmation.
3. I would also like the authors to clarify if the OxoScan-MS by itself can provide complete structures without the use of the Orbitrap LC-MS/MS and if not what are the limitations? Notably the OxoScan-MS provides 230 differential expressed “glycopeptide features” between healthy controls and severely affected patients but only 26 of these were selected for conventional glycoproteomics. Why? What was this selection based on? Where is the limitation – conceptual planning, sensitivity, time or instrumentation? How many of the 230 features could be identified as well-defined glycopeptides from the pooled material using the Orbitrap? Using the OxoScan-MS?
4. I did not note any oxonium ions provided by fragmentation of glycans containing fucose on the antennae – was this deliberately left out due to technical difficulties or was there any other reason? Changes in fucosylation as well as in sialylation is a typical character of glycoproteins in cancer and sialylated fucosylated structures (e.g. S_{Lex} or S_{Lea}) are ligands involved in inflammatory responses through the selectins so this limitation should also be commented on in the Discussion. Additionally, a recent study showed that lack of core fucosylation of IgG was increased in severely ill COVID-19 patients (M. D. Larsen et al., *Science* 371, eabc8378 (2021). DOI: 10.1126/science.abc8378), a finding the authors should search for and comment on.

5. If glycan oxonium ions, additional to the ones now used, may easily be added to the software then this could be lifted as a strength of the method and could also help the field in future determination of glycan structures. Does the OxoScan-MS method, allow for differentiating e.g. GlcNAc and GalNAc isomers or NeuAc glycosidic linkage positions similar to what has now been reported repeatedly in the literature , e.g. Ref #49 and Pett et al. Effective Assignment of α 2,3/ α 2,6-Sialic Acid Isomers by LC-MS/MS-Based Glycoproteomics. *Angew Chem Int Ed Engl.* 2018 and reviewed in Chernykh A, Kawahara R, Thaysen-Andersen M. Towards structure-focused glycoproteomics. *Biochem Soc Trans.* 2021?
6. Did the authors use any oxonium ion m/z characteristics for identifying bisecting HexNAc? There are, in Table S5 and Fig S4 some N5H5 glycoforms that might contain bisecting GlcNAc but do the authors know if these glycans carry bisecting residues? If so how were these assessed?
7. Did the authors use any oxonium ion m/z characteristics for identifying HexNAc-HexNAc glycans??
8. Most glycoproteins in human plasma are indeed N-glycosylated and the differential changes for the 26 glycoproteins analyzed in COVID-19 patients and plotted in Fig S3 are all N-glycosylated except for AHSG, Ser346 N1H1S1 which was obviously O-glycosylated and differentially expressed (Fig 3d). AHSG glycopeptides also appeared as N-glycopeptides (Fig S2). I would urge the authors to a comment on the relative quantification of N- vs O-glycopeptides of this – or any other - glycoprotein in plasma of diseased patients and controls. Is the method skewed towards detecting N-glycopeptides rather than equally well detecting N- and O-glycopeptides??
9. Finally I would like to comment that we could not open the MS raw files available at the website although the log-in to this site was ok.
10. Minor comments; the N-glycan of Fig 1 is not linked to the correct amino acid of the IgG peptides.
11. Minor comments: Ref 56, journal information is missing.

Reviewer #3 (Report for the authors (Required)):

The manuscript “OxoScan-MS: Oxonium ion scanning mass spectrometry facilitates plasma glycoproteomics in large scale by White et al. is a well written description of a new method to rapidly detect and provide relative quantification of glycopeptides. The method combines scanning sequential window acquisition of all theoretical mass spectra (SWATH-MS) with a technique they call OxoScan-MS which includes screening for oxonium ions to screen for glycopeptides. The manuscript is very well written and of high technical quality. It was a pleasure to read, and I only have a few minor suggestions listed below. The technique scanning SWATH was developed by this group and published by Messner et al. in *Nature Biotechnology*. Scanning SWATH is a technical breakthrough. The data provided in this manuscript is impressive and the addition of oxonium ion screening to the scanning SWATH provide a wealth of data on glycosylation difference in COVID-19 patient form controls. Oxonium ion screening is a fairly common technique and the addition of it in the form of OxoScan is a nice addition to the method. This addition may not be novel enough for *Nature Biotechnology*, however and may be better suited for a journal such as *Analytical Chemistry* of the *Journal of the American Society for Mass Spectrometry*.

Minor Comments

1. The number of COVID-19 and control patients should also be stated in the abstract.
2. HCD-pd-ETD needs to be defined the first time it is used.
3. The term feature is common for metabolomics but less common in proteomics. The authors may want to consider defining feature earlier than page 7.
4. On figure 1f -- the glycan is one residue to the left of the N and is on the column with the F and not the N. Also – the yellow text is really hard to read. Perhaps a dark blue or dark green would be better than yellow.
5. Line 160-168 – The authors claim to match their features to those from a MALDI TOF manuscript. The

authors will need to explain why this is a valid approach and present data that can validate this approach. After reading Wieczorek et al., it is not clear how the authors used the MALDI TOF data to convincingly assign their features. Additionally, the Wieczorek paper did not go to any additional effort to confirm the structural assignments of their mass spectral peaks, so it is not clear why their assignments were used in this manuscript. The data is the supplemental using the approach of Ang et al. was more convincing. Also - it should be more clearly stated if the authors also used the stepped-energy higher energy collisional dissociation approach of Chandler et al. to characterize the array of glycans.

6. The figure 1 legend needs to be more descriptive. These are complicated figures; it would be much easier for the reader to be walked through them by the authors.

Thu 02 Mar 2023

Decision on Article NBME-22-1691A

Dear Prof Ralser,

Thank you for your revised manuscript, "OxoScan-MS: Oxonium ion scanning mass spectrometry for plasma glycoproteomics at large scale". Having consulted with the original reviewers (whose comments you will find at the end of this message), I am pleased to write that we shall be happy to publish the manuscript in *Nature Biomedical Engineering*.

We will be performing detailed checks on your manuscript, and in due course will send you a checklist detailing our editorial and formatting requirements. You will need to follow these instructions before you upload the final manuscript files. In the meantime, please consider the below minor recommendations from the reviewers.

Best wishes,

Pep

Pep Pàmies
Chief Editor, Nature Biomedical Engineering

Reviewer #1 (Report for the authors (Required)):

White et al. present a new data-independent glycoproteomic method - OxoScan-MS. The methodology is an effective alternative to the current DIA methodology for glycosylation analysis. Although oxonium ions are the glycopeptide fragments with the lowest selectivity, the authors confirmed that the methodology's performance has promising results, evaluated by orthogonal measurement by specific PRM analysis. The revised manuscript is significantly improved compared to the first submission, and I recommend that this manuscript be published in Nature Biomedical Engineering after minor revision.

The authors have addressed all of the reviewers' concerns, and I have just one minor comment. The authors should state in the manuscript that the oxonium outer arm fucosylation "specific" ions could be a fucose rearrangement product and that the specificity of these ions was not confirmed by structure-specific fucosylation analysis such as exoglycosidase digestion or glycostructure-specific separation.

Reviewer #2 (Report for the authors (Required)):

Relating to the originally submitted manuscript and the serious responses from the authors to reviewers' comments I feel very satisfied with the revised manuscript. It has been greatly improved and I am happy to recommend publication after the following VERY minor corrections of revisions.

1. In the Figure legend of Fig S2 there is the "m/z" specification missing within the brackets for the oxonium ions. (I was also happy to realize that the last sentence of this figure legend found in the response letter is deleted from the final figure legend.)

2. There is a misconception slowly being introduced in the literature concerning the abbreviation for the histo-blood group antigens of the Lewis system; Lewis a should be abbreviated "Lea" with the "a" raised and in lower case. Similarly Lewis x should be abbreviated "Lex" with the "x" raised and in lower case. Consequently Sialyl Lewis a should be abbreviated "SLea" and Sialyl Lewis x abbreviated "SLeX" with the "a" and "x" raised and in lower case.

Reviewer #3 (Report for the authors (Required)):

The manuscript by Matthew White et al. "OxoScan-MS: Oxonium ion scanning mass spectrometry for plasma glycoproteomics at large scale" is well written and was thoroughly reviewed by three independent reviewers. I found the manuscript well written and very complete. The authors completely answered the reviewer comments and made appropriate changes to the manuscript. I feel that the paper is now ready for publication.

On a side note, I found the level of information contained in this manuscript by the oxoscan method to be compelling. It may be nice to someday compare the results to the less specific NMR analysis for "GlycA" that is a subject of many publications such as Otvos J.D., Guyton J.R., Connelly M.A., Akapame S., Bittner V., Kopecky S.L., Lacy M., Marcovina S.M., Muhlestein J.B., Boden W.E. Relations of GlycA and lipoprotein particle subspecies with cardiovascular events and mortality: A post hoc analysis of the AIM-HIGH trial. *J. Clin. Lipidol.* 2018;12:348–355. doi: 10.1016/j.jacl.2018.01.002. and Fuertes-Martín et al in *J. Clin Med.* 2020 Jan 27;9(2):354. doi: 10.3390/jcm9020354.

Rebuttal 1

Response to Reviewers, White et al.

We thank all three reviewers for their positive evaluation and we highly appreciate their time and effort and the detailed feedback. We found the comments very constructive and have addressed them one-by-one in the response letter below. We have incorporated a number of further analyses as suggested and believe that the revisions have substantially improved the quality and clarity of the manuscript.

Point-by-point reply

1. Joint reply to comments made by all reviewers

Reviewers #1 and #2 both suggested further clarification on the possibility of applying OxoScan-MS for structure-specific oxonium ions, including antennary fucosylation, SLeX/SLeA and LacdiNAc structures. We agree that such an analysis is highly relevant in developing clinical biomarkers. We would like to emphasise that OxoScan-MS is inherently ion-agnostic (in the sense any fragment ions in the acquired range can be extracted) and therefore other informative ions can be included in the analysis, either in conjunction with reported oxonium ions or individually. This represents a further strength of the method.

To specifically address the reviewers' questions, we re-analysed a plasma sample acquired by OxoScan-MS, including other structure-specific oxonium ions as suggested, specifically HexNAc-HexNAc (m/z 407.165, including LacdiNAc), HexNAc-Hex-Fuc (m/z 512.197, including LeX/LeA) and HexNAc-Hex-Fuc-Neu5Ac (m/z 803.293, including SLeX/SLeA). While these three structure-specific oxonium ions are less abundant than the originally reported oxonium ions, they all clearly display features across the oxonium ion maps (Figure S2a in the revised manuscript, reproduced as Figure R1a below). Furthermore, as structure-specific oxonium ions are inherently non-ubiquitous, lower overall abundance is to be expected.

We further found overlap between the localisation of structure-specific ions and common oxonium ions, shown by a representative example of HexNAc-HexNAc overlaying with Hex- and HexNAc- but not Neu5Ac-derived oxonium ions (Figure S2b in the revised manuscript, included as Figure R1b below). Although we have not performed full glycopeptide sequence identification, the oxonium ion profiles suggest this is a non-sialylated glycopeptide with a HexNAc-HexNAc motif and further demonstrates the flexibility of OxoScan to be used with non-standard ions dependent on the application. We do note that although the extracted masses correspond to specific structures (e.g. SLeX/SLeA for m/z 803.293), they do not distinguish between linkage or glycan isomers. This is a common difficulty in glycoproteomics, which Reviewer #2 correctly points out has been predominantly addressed through highly targeted (MSn) analyses^{1,2}. As a result, distinguishing e.g. GalNAc/GlcNAc or α 2,3/ α 2,6-sialic acid linkages is currently beyond the scope of OxoScan-MS.

Nevertheless, specific methods optimised for structurally-informative oxonium ions could be applied for further insight. We have included this analysis in the Supplementary Information (Figure S2), Results section (p.8/9), and in the discussion as follows:

“Thus, although linkage- and structure-specific information can be gleaned from glycopeptide MS/MS spectra (including structure-specific fragment ions and ion-ion ratios²⁻⁴), our analysis is restricted to the monosaccharide compositions reported by two widely used glycopeptide assignment tools (MSFragger-Glyco/Byonic). We want to emphasise however, that OxoScan-MS data can be

retrospectively mined for custom fragment ions of interest, including structure-specific oxonium ions and can therefore be easily integrated with future developments in applying non-ubiquitous oxonium ions or fragment ion ratios for glycan classification, including those relating to clinically-relevant glycan structures, such as Lewis A/X epitopes, rationally designed chemical probes or other endogenous PTMs.⁵⁻¹⁰

Figure R1 (Figure S2 in the revised manuscript): OxoScan-MS allows for retrospective extraction of custom (e.g. structure-specific) ions of interest. a. Full width 2-dimensional oxonium ion maps for a plasma sample measured in the COVID-19 cohort, with 3 common oxonium ions (186.076, 204.087, 274.092) from single monosaccharide units and more specific ions corresponding to 2-4 saccharide units (N = HexNAc, H = Hex, S = Neu5Ac, F = Fucose). **b.** 2-dimensional oxonium ion maps for a plasma sample measured in the COVID-19 cohort. The HexNAc-HexNAc ion (m/z 407.165) matches with oxonium ions derived from HexNAc and Hex, but not Neu5Ac.

2. Individual point-by-point replies to all reviewer comments

Reviewer #1 (Report for the authors (Required)):

White et al. present a new data-independent glycoproteomics method - OxoScan-MS. This methodology is another application in the growing field of data-independent glycoproteomics. OxoScan was developed on the analysis of isolated IgG glycopeptides from human plasma. The methodology shows promising performance, but oxonium ions are the glycopeptide fragments with the lowest selectivity, so the performance of the methodology for low-represented glycopeptides in high-abundance background remains questionable and needs further testing. The authors tested the methodology on plasma samples from a cohort hospitalized for COVID-19. Over all, this is a valid piece of glycoproteomics work and I recommend that this manuscript be published in a Nature Biomedical Engineering journal after major revision.

We thank the reviewer for their favourable assessment of our work's value for the glycoproteomics field.

I just have a few questions: The authors claim to have obtained quantitative results for more than 1k glycopeptides. For the above reason, I recommend verifying the results of significantly changed glycopeptides with a more specific PRM methodology.

We thank the reviewer for their comment. We agree that further verification by targeted MS would validate our OxoScan-MS approach. We therefore generated a high-resolution multiple reaction monitoring (MRM-HR) method on a Sciex ZenoTOF 7600, targeting 24 glycopeptide precursors, as

well as 40 unmodified (10 for internal reference and 30 adjacent/non-glycosylated) precursors. Of these targets, 17 of 24 glycopeptides passed all assignment and quality criteria. We applied this method to the COVID-19 cohort. Please refer to the updated Methods section (p21) for further details.

Encouragingly, we found excellent agreement in MS/MS spectra between OxoScan and MRM-HR, for example, when comparing for the N4H5S2 N630 glycopeptide from transferrin (Figure R2a). Furthermore, the fold-changes of the glycopeptides passing our quality criteria (please see Methods) are highly correlated ($\rho = 0.863$) across the clinical cohort between OxoScan and MRM-HR (Figure R2b). We further find that the intensities of ubiquitous oxonium ions and glycopeptide-specific (Y-type) ions showed excellent agreement (Figure R2c). We have added these findings to the Results section of the revised manuscript (p.14/15) and updated Figure 5 accordingly.

Figure R2 (Figure 5b-d in the revised manuscript): Validation of OxoScan-MS by targeted mass spectrometry (HR-MRM). **a**) Back-to-back MS/MS spectra of the TF N4H5S2 Asn630 glycopeptide as acquired from OxoScan-MS (Sciex TripleTOF 6600) and MRM-HR (Sciex ZenoTOF 7600). **b.** Scatter plot showing correlation of 17 glycopeptides quantified with OxoScan-MS (y-axis) and MRM-HR (x-axis). **c.** Quantification of glycopeptides from MRM-HR with either: summed oxonium ions (y-axis) or glycopeptide-specific Y-type ions (x-axis) show excellent agreement. For **b** and **c**, axes are log₁₀-transformed and the Spearman correlation coefficient (ρ) is shown on the plot.

a. How did the authors work with less abundant but more structured informative oxonium ions such as 407 (LacdiNAc etc.)? Do the authors plan to extend the methodology for site- and structure-specific data-independent glycoproteomics?

We thank the reviewer for their insightful question. We have addressed this in our joint reply to all reviewers (p.1 of this reply letter).

b. The authors analyzed protein abundance in addition to glycopeptide abundance to normalize the data. How do the authors address glycosite occupancy, could this information be provided in Figure 5?

We thank the reviewer for the stimulating question. We re-measured OxoScan-prioritised glycopeptides with MRM-HR of both the glycopeptide and non-modified peptides from the same protein, and expanded Figure 5 to better reflect on this aspect. We find examples of abundance changes of glycopeptides being highly correlated with that of their corresponding adjacent (i.e. from the same protein, but different sequence) peptides, indicating that their glycosylation status is not differentially regulated (Figure 5e in the revised manuscript, HP + TF panels) relative to protein abundance changes. Other examples indicate differential glycosylation, where glycopeptides change differently than the corresponding adjacent peptides with increasing disease severity (Figure 5e in the revised manuscript, AHSG, ORM1, IGHA1;IGHA2 panels, for a scheme explaining the principle, please see the new Fig. S7)).

We further included non-glycosylated peptides that had been detected in peptide-level analyses previously¹¹. Interestingly, for AHSG S346 and TF N630, the non-glycosylated peptidiform abundance changed significantly compared to aggregated adjacent peptides (Figure S6c), indicating a change in glycosylation occupancy with disease severity. As with all peptide-level inference, care must be taken to account for the range of potential PTMs in a given peptide sequence, especially from a loss-of-signal¹², however we demonstrate here the potential insights afforded by parallel peptide and glycopeptide quantification.

We have added details and an explanatory figure of the use of glycopeptides, adjacent and non-modified peptides (Table S6, Figure S6) and discussion of these findings to the Results section in the revised manuscript (p.14/15, Figure 5e).

c. In Figure 5, panel c shows the IgA1 and IgA2 shared glycopeptide changes, but which form of IgA are the protein changes

We thank the reviewer for bringing this to our attention. Whenever peptide sequences with a given glycosylation site were shared between two isoforms (e.g. IGHA1;IGHA2 example in Figure 5a/e), only shared peptides were used for quantification. It is worth noting that in the case of IgA, there is no significant difference between the isoforms on the protein level and we see similar trends when normalising by peptides unique for either IGHA1, IGHA2 or IGHA1;IGHA2. We have added a clarifying comment to the figure legend in the revised manuscript describing the normalised plots (p15) and added a Supplementary Figure showing the protein group normalisations (Figure S6b, reproduced below as Fig. R3).

Figure R3 (Figure S6b in the revised manuscript): IGHA1;IGHA2 shared glycopeptides show similar trends when normalised to either IGHA1-specific (red), IGHA2-specific (blue) or IGHA1;IGHA2 shared (green) peptides. Boxplots showing ratios of 3 IgA glycopeptides normalised to peptides specific to each IgA isoform or shared between isoforms. Similar trends are seen for all normalisations.

Reviewer #2 (Report for the authors (Required)):

This well written manuscript describes the development, validation and application of a new strategy, OxoScan-MS, for unbiased large-scale identification and relative quantification of tryptic glycopeptides, obtained without enrichment steps, from glycoproteins of biological fluids e.g. plasma or serum, within a short time frame suitable for patient cohort studies performed either at university core or clinical reference laboratories.

The strategy is built on the already well-known characteristic positive – oxonium - ion fragmentation of glycans obtained from MS/MS fragmentation through CID and HCD, but it is here taken to a systematic screening level with DIA-SWATH-MS using fast LC C18 NanoAcquity chromatography and a Sciex triple TOF 6600 instrument and a newly developed, now freely available software OxoScan, to create 2D Oxonium ion profiles containing thousands of identifiable glycopeptides from a single sample. For complete structural identification of glycopeptides pooled samples were analyzed on another nLC-MS/MS instrument (Ultimate 300 RSLCnano linked to an Orbitrap Eclipse run in HCD-pd-ETD) and data were interpreted using the Byonic software.

After validation of the method on standard protein mixtures and plasma samples the OxoScan-MS was applied to analyze glycopeptides of 30 plasma samples of patients with mild to severe COVID-19 versus 15 healthy controls. Among 1.102 unique glycopeptide features in all samples, 90 % (1.002) were quantifiable across all clinical samples and spanned over four orders of magnitude. Data were bioinformatically digested to heatmaps for each individual and each glycan feature as well as with Principal Component analysis and also compared with proteomic analyses of the same samples giving an appreciation of which of 26 selected glycoproteins that significantly changed their concentrations (e.g. alpha-2-HS-glycoprotein and hemopexin) or which changed their site-specific glycosylation profiles (e.g. IgA, alpha-1-antitrypsin as well as fibrinogen alpha and beta chains) related to the (stage of) COVID-19 disease. Although complex patterns and individual variabilities were obtained, clear tendencies separating healthy from diseased individuals were revealed and are worthy of further studies.

The OxoScan-MS method shows great promise and will most certainly be welcomed as a useful tool in the search for complex patterns of disease biomarkers carried on glycoproteins of body fluids and of cultured cell and tissue media.

I have only a few mainly technical comments for the authors.

We thank the Reviewer for their very positive assessment of our study, and the potential of OxoScan-MS.

1. I think it is important to point out, as a limitation of the study, that the structural data provided are giving monosaccharide compositions of glycans rather than glycan structures. The authors use the term “features” – which as some early point in the text should be explained in structural terms.

We agree, and apologise if this limitation was not made sufficiently clear. We have revised the text throughout and use ‘glycopeptide features’ wherever appropriate. Moreover we added clarification in the results section first describing glycopeptide identification and mapping to glycopeptide features as follows:

“It is worth noting that both Byonic and MSFragger provide assignment of glycan compositions, but do not inform on linkage- or structure-specific glycan characteristics. As such, the glycan identity assigned to a given glycopeptide feature reflects the monosaccharide composition, as opposed to specific structural assignment.”

2. In this respect is also important to discuss and reference in the text (preferentially in the Discussion) the weakness of the Byonic software in interpreting isobaric m/z peaks into composition or even structures (See ref Kawahara R, et al Community evaluation of glycoproteomics informatics solutions reveals high-performance search strategies for serum glycopeptide analysis. Nat Methods. 2021 Nov;18(11):1304-1316. doi: 10.1038/s41592-021-01309-x. Epub 2021 Nov 1. Erratum in: Nat

Methods. 2021 Dec 10; Erratum in: Nat Methods. 2022 Jan;28(1):214. PMID: 34725484; PMCID: PMC8566223.).

We thank the reviewer for their helpful suggestion. We have added the references and a discussion about both tools (and their agreement/differences) in the results section as follows below:

“Recent studies have shown glycoproteomic assignment can vary significantly with the analysis software and settings,¹³ so we performed glycopeptide identification with both Byonic¹⁴ (Protein Metrics Inc.) and MSFragger-Glyco¹⁵ and further filtered post-processing for assignment quality (see Methods - DDA Data Processing).”

For solving this issue either manual interpretation with well defined criteria or alternative software should be used for structural confirmation.

As suggested by the reviewer, we validated glycopeptide assignments from Byonic by orthogonal searching with MSFragger-Glyco.¹⁵ The orthogonal software tools agreed on 22 of the 26 identified glycopeptides. We applied this as a further filter of assignment quality and kept only the shared assignments in the revised manuscript. Results (p11/12), Figures (Figure 4, 5) and Discussion (p17) have been updated accordingly.

3. I would also like the authors to clarify if the OxoScan-MS by itself can provide complete structures without the use of the Orbitrap LC-MS/MS and if not what are the limitations?

We apologise if this has not been made sufficiently clear. We see the strength, and main application of OxoScan-MS at the screening stage of a glycoproteomic analysis, specifically in large sample series such as plasma sample cohorts. The screening of oxonium ions by exploiting a Q1 scanning dimension proved highly sensitive (i.e. does not require glycopeptide enrichment and works on standard tryptic digests) and quantitative, but it can only provide the amount of information about the glycan that is derived from the diagnostic oxonium ions. In our study we combine it therefore with a gold-standard glycopeptide identification strategy on an Orbitrap instrument (HCD-pd-ETD) for obtaining more detailed information about glycopeptide composition. Certainly, the identification could equally be conducted on another platform. To us, the most attractive option is combining both quantitation by OxoScan-MS and glycopeptide identification on one platform, e.g. on the recently released Sciex ZenoTOF 7600, which offers electron-transfer based fragmentation capabilities. We have included this point to the discussion as follows:

“We further note that while different LC-MS platforms were used for glycopeptide quantification and identification as a proof-of-concept, next-generation mass spectrometers which integrate both scanning quadrupole capability and multiple complementary fragmentation strategies amenable for glycopeptide analysis could significantly streamline the reported approach.”

Notably the OxoScan-MS provides 230 differential expressed “glycopeptide features” between healthy controls and severely affected patients but only 26 of these were selected for conventional glycoproteomics. Why? What was this selection based on? Where is the limitation – conceptual planning, sensitivity, time or instrumentation? How many of the 230 features could be identified as well-defined glycopeptides from the pooled material using the Orbitrap? Using the OxoScan-MS?

We thank the reviewer for the questions and apologise that this was not sufficiently clear. We followed a heuristic filtering approach to validate a subset of our hits detected in our screening. In the first filtering step, we compared glycopeptide feature matches between OxoScan-MS and HCD-pd-

ETD-Orbitrap-DDA data. Of the 1,002 glycopeptide features consistently quantified across the cohort, 167 had putative matches based on scaled retention times ($\pm 20\%$) and precursor m/z values (± 1 Th) between OxoScan / Orbitrap DDA platforms (including only glycopeptides identified by both Byonic and MSFragger-Glyco). Considering only the 230 differentially abundant glycopeptide features in the COVID-19 cohort, the number of putative matches dropped to 71.

We then applied the described assignment quality criteria and manually validated precursor matching using high-resolution TOF-MS measurements and comparison of MS/MS fragmentation spectra between OxoScan/DDA data (as shown in Figure 4 in the revised manuscript), initially focusing on highly abundant glycopeptide features with good MS1 signal intensity, as a proof-of-principle of OxoScan/DDA matching. A further filtering step was applied at this stage, where any precursor with a co-eluting glycopeptide within 5 m/z (distinguished by a narrow-window OxoScan method) was removed. Having confirmed that OxoScan features could be confidently matched to DDA-reported glycopeptide assignments, we expanded to include other reported glycopeptides from the proteins identified in the first round, as well as features with high fold-changes between disease severities. The resulting number of glycopeptides matched is therefore due to the application of stringent quality filters, overlap between OxoScan/DDA data and manual validation of matches.

We would further note here that while we applied two different acquisition strategies, LC regimes and MS instruments in our proof-of-principle, we anticipate that matching OxoScan features to glycopeptide identifications will be significantly streamlined by next-generation instruments capable of both scanning-quadrupole acquisition and fragmentation modes optimal for glycopeptide identification. We have added these points to the Discussion (p17) and Methods (p21-23).

4. I did not note any oxonium ions provided by fragmentation of glycans containing fucose on the antennae – was this deliberately left out due to technical difficulties or was there any other reason? Changes in fucosylation as well as in sialylation is a typical character of glycoproteins in cancer and sialylated fucosylated structures (e.g. SLex or SLea) are ligands involved in inflammatory responses through the selectins so this limitation should also be commented on in the Discussion.

We thank the reviewer for the insightful question, and this comment goes in a similar direction of a question by Reviewer #1. We have addressed it in the common response section at the top of the response letter.

Additionally, a recent study showed that lack of core fucosylation of IgG was increased in severely ill COVID-19 patients (M. D. Larsen et al., *Science* 371, eabc8378 (2021). DOI: [10.1126/science.abc8378](https://doi.org/10.1126/science.abc8378)), a finding the authors should search for and comment on.

We thank the reviewer for this suggestion - however the mentioned study primarily focuses on antibodies specific to either the SARS-CoV-2 spike (anti-S) or nucleocapsid (anti-N) proteins and therefore are not necessarily comparable to profiles of total IgG pools. OxoScan-MS can be adapted for quantification of IgG glycoforms in enriched samples or neat plasma, however we focused on a plasma proteomic discovery approach in our current proof-of-principle study. We have added this to the discussion (p17).

5. If glycan oxonium ions, additional to the ones now used, may easily be added to the software then this could be lifted as a strength of the method and could also help the field in future determination of

glycan structures. Does the OxoScan-MS method, allow for differentiating e.g. GlcNAc and GalNAc isomers or NeuAc glycosidic linkage positions similar to what has now been reported repeatedly in the literature , e.g. Ref #49 and Pett et al. Effective Assignment of α 2,3/ α 2,6-Sialic Acid Isomers by LC-MS/MS-Based Glycoproteomics. Angew Chem Int Ed Engl. 2018 and reviewed in Chernykh A, Kawahara R, Thaysen-Andersen M. Towards structure-focused glycoproteomics. Biochem Soc Trans. 2021?

We agree with the reviewer that adding further oxonium ions easily into the OxoScan-MS approach is a key strength of the method. We address the comments of both Reviewer #1 and this reviewer regarding structure-specific oxonium ions in the section addressed to multiple reviewers (Page 1 of this response letter).

6. Did the authors use any oxonium ion m/z characteristics for identifying bisecting HexNAc? There are, in Table S5 and Fig S4 some N5H5 glycoforms that might contain bisecting GlcNAc but do the authors know if these glycans carry bisecting residues? If so how were these assessed?

The reviewer makes a good point. To our knowledge, the most convincing ‘diagnostic ions’ for bisecting HexNAc typically have been found with HCD and represent characteristic fragmentation of the glycan with intact peptide (e.g. Pep+HexNAc3Hex1¹⁶). As these cannot be extracted ubiquitously like common oxonium ions without previously knowing the glycopeptide identity, we have not included them in our study.

Regarding the N5H5 glycoform, the reviewer is indeed correct that this could contain bisecting HexNAc. As mentioned in response to Reviewer #1 and above, we have clarified that in accordance with the glycopeptide assignment tools used (MSFragger-Glyco and Byonic) and the complexity of subsequent validation of these assignments in CID fragmentation, we report glycan compositions.

This clearly represents a limitation of our approach (and other common glycoproteomics approaches too) and we have added this into the discussion (p16/17), as detailed in our response to Reviewer #2’s first question.

7. Did the authors use any oxonium ion m/z characteristics for identifying HexNAc-HexNAc glycans??

We thank the reviewer for the insightful question and have addressed it in a response to multiple reviewers at the top of the response letter.

8. Most glycoproteins in human plasma are indeed N-glycosylated and the differential changes for the 26 glycoproteins analyzed in COVID-19 patients and plotted in Fig S3 are all N-glycosylated except for AHSG, Ser346 NIH1S1 which was obviously O-glycosylated and differentially expressed (Fig 3d). AHSG glycopeptides also appeared as N-glycopeptides (Fig S2). I would urge the authors to a comment on the relative quantification of N- vs O-glycopeptides of this – or any other - glycoprotein in plasma of diseased patients and controls. Is the method skewed towards detecting N-glycopeptides rather than equally well detecting N- and O-glycopeptides??

The reviewer makes an excellent point. The COVID OxoScan-MS method indeed was optimised for N-glycopeptides and we further observed in DDA data of non-enriched serum that 82% of assigned PSMs after filtering represented N-glycopeptides. Taking into account both factors, it is not surprising

that we identify predominantly N-glycopeptides, especially given the extra difficulty in dealing with O-glycopeptides (e.g. the limitation of trypsin in separating multiple O-glycosylation sites¹⁷). We have added a comment in the discussion directly addressing this limitation, as follows:

“We also note that in the current study we identify predominantly N-glycopeptides, but future optimisation for O-glycan derived fragment ions and O-glycan enrichment strategies could improve the detection of O-glycosylated peptides. This is a common trade-off in plasma (glyco)proteomics experiments, however for our purposes we focused on increasing the practical throughput and reducing costs of glycoproteomics experiments, thus incorporating minimal extra handling steps.”

9. Finally I would like to comment that we could not open the MS raw files available at the website although the log-in to this site was ok.

We apologise to the reviewer for this issue. We have verified that the files on MassIVE are not corrupted. We find PeakView (AB Sciex, v2.2) works to visualise these files and Skyline should also be compatible. We have added a comment to the Data/code availability section (p24) about compatible software.

10. Minor comments; the N-glycan of Fig 1 is not linked to the correct amino acid of the IgG peptides.

We thank the reviewer for this comment, we have amended the error and updated Figure 1 in the revised manuscript.

11. Minor comments: Ref 56, journal information is missing.

This refers to a relevant poster presented at HUPO 2018 and available at https://www.waters.com/webassets/cms/library/docs/2018hupo_geethings_glycopeptide_fragmentation.pdf. We have added a note to the reference.

Reviewer #3 (Report for the authors (Required)):

The manuscript “OxoScan-MS: Oxonium ion scanning mass spectrometry facilitates plasma glycoproteomics in large scale by White et al. is a well written description of a new method to rapidly detect and provide relative quantification of glycopeptides. The method combines scanning sequential window acquisition of all theoretical mass spectra (SWATH-MS) with a technique they call OxoScan-MS which includes screening for oxonium ions to screen for glycopeptides. The manuscript is very well written and of high technical quality. It was a pleasure to read, and I only have a few minor suggestions listed below. The technique scanning SWATH was developed by this group and published by Messner et al. in Nature Biotechnology. Scanning SWATH is a technical breakthrough. The data provided in this manuscript is impressive and the addition of oxonium ion screening to the scanning SWATH provide a wealth of data on glycosylation difference in COVID-19 patient form controls. Oxonium ion screening is a fairly common technique and the addition of it in the form of OxoScan is a nice addition to the method. This addition may not be novel enough for Nature Biotechnology, however and may be better suited for a journal such as Analytical Chemistry of the Journal of the American Society for Mass Spectrometry.

Minor Comments

1. The number of COVID-19 and control patients should also be stated in the abstract.

We thank the reviewer for the helpful comment, cohort demographics have been added to the abstract as follows:

“We apply OxoScan-MS to profile the plasma glycoproteome in 30 COVID-19 patients and 15 healthy controls, consistently quantifying 1,002 glycopeptide features across the inpatient cohort.”

2. HCD-pd-ETD needs to be defined the first time it is used.

A definition of HCD-pd-ETD has been added when first mentioned in the introduction, as follows:

“...and utilised an orthogonal acquisition approach (higher-collisional dissociation with oxonium ion-dependent triggering of electron-transfer dissociation fragmentation, HCD-pd-ETD) to perform glycopeptide identification.”

3. The term feature is common for metabolomics but less common in proteomics. The authors may want to consider defining feature earlier than page 7.

We agree with the reviewer and have added the following comment when the concept of features is introduced in the first results section:

“It is worth noting that ‘glycopeptide features’ represent unique retention time-precursor m/z coordinates and are not unambiguously identified glycopeptides at the point of detection.”

4. On figure 1f -- the glycan is one residue to the left of the N and is on the column with the F and not the N. Also – the yellow text is really hard to read. Perhaps a dark blue or dark green would be better than yellow.

We thank the reviewer for the comment. We have amended the glycan position and changed the colour scheme to be more easily visible, see Figure 1f in the revised manuscript.

5. Line 160-168 – The authors claim to match their features to those from a MALDI TOF manuscript. The authors will need to explain why this is a valid approach and present data that can validate this approach. After reading Wieczorek et al., it is not clear how the authors used the MALDI TOF data to convincingly assign their features. Additionally, the Wieczorek paper did not go to any additional effort to confirm the structural assignments of their mass spectral peaks, so it is not clear why their assignments were used in this manuscript. The data is the supplemental using the approach of Ang et al. was more convincing.

We thank the reviewer for this point and agree that further comparison is useful. We have further validated the OxoScan IgG data with LC-MS based IgG glycopeptide profiling carried out by Momčilović et al. (*Anal. Chem.* 2020, 92, 6, 4518–4526). In their study, IgG glycopeptides are analysed by nanoLC-ESI-MS/MS and database searching with Byonic. Comparing to their assignments, we find all 30 IgG glycopeptides from OxoScan-MS match up with both the initial assignment from Wieczorek et al., as well as those assigned in Momčilović et al, with a maximum precursor mass error between our measurements and Momčilović et al. of 20.3 ppm (mean = 8.0 ppm, standard deviation = 5.1 ppm). This represents a standard mass accuracy tolerance for the instrument used (Sciex TripleTOF 6600). Furthermore, we observe identical retention time behaviour between IgG subclasses, with IgG1, IgG4 and IgG2/3 eluting sequentially, shown in Figure 1 of Momčilović et al. and Figure 1f of this study.

In conjunction with the MALDI-TOF analysis of IgG, we feel this confidently supports the assignments originally reported in this study. It is worth emphasising that we report only glycan compositions. We have added the matching of nanoLC-MS identified features to Table S1 and all matching parameters, shown below, and included this in the Results section (p5).

Table S1 in the revised manuscript: IgG glycopeptides identified by OxoScan-MS analysis.

IgG	Glycan	m/z	Monoisotopic Mass	Charge	RT	[M+H] ⁺	Peptide Mass	Glycan Mass	peak_num	Literature Precursor Monoisotopic	PPM Mass Error
1	G0	830.0054	2486.9943	3	27.55	2488.0016	1188.5047	1299.4969	17	2486.981	5.484
1	G0F	878.6832	2633.0277	3	26.85	2634.035	1188.5047	1445.5303	7	2633.039	4.128
1	G0FN	946.3678	2836.0815	3	27.95	2837.0888	1188.5047	1648.5841	14	2836.118	12.849
1	G1F	932.6988	2795.0745	3	26.55	2796.0818	1188.5047	1607.5771	4	2795.091	6.044
1	G1F	932.7058	2795.0955	3	26.96	2796.1028	1188.5047	1607.5981	18	2795.091	1.47
1	G1FN	1000.3946	2998.1619	3	27.75	2999.1692	1188.5047	1810.6645	13	2998.171	2.957
1	G1FS1	1029.7458	3086.2155	3	28.75	3087.2228	1188.5047	1898.7181	21	3086.187	9.296
1	G2F	986.7129	2957.1168	3	26.46	2958.1241	1188.5047	1769.6194	8	2957.144	9.271
1	G2FN	1054.4152	3160.2237	3	27.55	3161.231	1188.5047	1972.7263	20	3160.224	0.035
1	G2FS1	1083.7589	3248.2548	3	28.15	3249.2621	1188.5047	2060.7574	11	3248.24	4.669
2	G0	819.3432	2455.0077	3	52.15	2456.015	1156.5149	1299.5001	35	2454.991	6.859
2	G0F	868.0212	2601.0417	3	50.95	2602.049	1156.5149	1445.5341	1	2601.049	2.718
2	G0FN	935.703	2804.0871	3	51.25	2805.0944	1156.5149	1648.5795	9	2804.128	14.636
2	G1F	922.0331	2763.0774	3	50.15	2764.0847	1156.5149	1607.5698	3	2763.102	8.756
2	G1F	922.031	2763.0711	3	50.75	2764.0784	1156.5149	1607.5635	5	2763.102	11.036
2	G1FN	989.7264	2966.1573	3	50.55	2967.1646	1156.5149	1810.6497	12	2966.181	7.978
2	G1FN	989.7217	2966.1432	3	50.95	2967.1505	1156.5149	1810.6356	27	2966.181	12.732
2	G1FS1	1019.0768	3054.2085	3	53.65	3055.2158	1156.5149	1898.7009	10	3054.197	3.762
2	G2F	976.0541	2925.1404	3	49.85	2926.1477	1156.5149	1769.6328	2	2925.154	4.792
2	G2FN	1043.7621	3128.2644	3	50.25	3129.2717	1156.5149	1972.7568	15	3128.234	9.785
2	G2FS1	1073.0854	3216.2343	3	52.75	3217.2416	1156.5149	2060.7267	6	3216.25	4.829
4	G0F	873.36	2617.0581	3	38.05	2618.0654	1172.5098	1445.5556	19	2617.044	5.514
4	G0FN	941.0292	2820.0657	3	38.55	2821.073	1172.5098	1648.5632	33	2820.123	20.333
4	G1F	927.3766	2779.1079	3	37.45	2780.1152	1172.5098	1607.6054	23	2779.096	4.105
4	G1F	927.3892	2779.1457	3	37.15	2780.153	1172.5098	1607.6432	69	2779.096	17.706
4	G1F	927.3757	2779.1052	3	37.95	2780.1125	1172.5098	1607.6027	43	2779.096	3.133
4	G1FN	995.058	2982.1521	3	38.15	2983.1594	1172.5098	1810.6496	36	2982.176	7.969
4	G2F	981.38	2941.1181	3	37.24	2942.1254	1172.5098	1769.6156	26	2941.149	10.614
4	G2FN	1049.0927	3144.2562	3	37.85	3145.2635	1172.5098	1972.7537	81	3144.229	8.75
4	G2FS1	1078.4419	3232.3038	3	39.55	3233.3111	1172.5098	2060.8013	24	3232.245	18.274

Also - it should be more clearly stated if the authors also used the stepped-energy higher energy collisional dissociation approach of Chandler et al. to characterize the array of glycans.

No stepped collision energy approach was applied in this study, instead a single rolling collision energy is applied which calculates the collision energy based on the *m/z* value of the centre of the precursor quadrupole (please refer to the Methods section).

6. The figure 1 legend needs to be more descriptive. These are complicated figures; it would be much easier for the reader to be walked through them by the authors.

We have added further explanation to the Figure 1 legend as suggested by the reviewer (p6). We further included the following note: “Also find an in-depth explanation of the scanning quadrupole concept and Q1 traces in Messner et al., 2021¹⁸”.

Supplementary References

1. Halim, A. *et al.* Assignment of saccharide identities through analysis of oxonium ion fragmentation profiles in LC-MS/MS of glycopeptides. *J. Proteome Res.* **13**, 6024–6032 (2014).
2. Chernykh, A., Kawahara, R. & Thaysen-Andersen, M. Towards structure-focused glycoproteomics. *Biochem. Soc. Trans.* **49**, 161–186 (2021).
3. Toghi Eshghi, S. *et al.* Classification of Tandem Mass Spectra for Identification of N- and O-linked Glycopeptides. *Sci. Rep.* **6**, 37189 (2016).
4. Pett, C. *et al.* Effective Assignment of α 2,3/ α 2,6-Sialic Acid Isomers by LC-MS/MS-Based Glycoproteomics. *Angew. Chem. Int. Ed Engl.* **57**, 9320–9324 (2018).
5. Cohen, E. N. *et al.* Elevated serum levels of sialyl Lewis X (sLeX) and inflammatory mediators in patients with breast cancer. *Breast Cancer Res. Treat.* **176**, 545–556 (2019).
6. Smith, B. A. H. & Bertozzi, C. R. The clinical impact of glycobiology: targeting selectins, Siglecs and mammalian glycans. *Nat. Rev. Drug Discov.* **20**, 217–243 (2021).
7. Stowell, S. R., Ju, T. & Cummings, R. D. Protein glycosylation in cancer. *Annu. Rev. Pathol.* **10**, 473–510 (2015).
8. Everley, R. A., Huttlin, E. L., Erickson, A. R., Beausoleil, S. A. & Gygi, S. P. Neutral Loss Is a Very Common Occurrence in Phosphotyrosine-Containing Peptides Labeled with Isobaric Tags. *J. Proteome Res.* **16**, 1069–1076 (2017).
9. Kelstrup, C. D., Frese, C., Heck, A. J. R., Olsen, J. V. & Nielsen, M. L. Analytical utility of mass spectral binning in proteomic experiments by SPectral Immonium Ion Detection (SPIID). *Mol. Cell. Proteomics* **13**, 1914–1924 (2014).
10. Calle, B. *et al.* Benefits of Chemical Sugar Modifications Introduced by Click Chemistry for Glycoproteomic Analyses. *J. Am. Soc. Mass Spectrom.* **32**, 2366–2375 (2021).
11. Messner, C. B. *et al.* Ultra-High-Throughput Clinical Proteomics Reveals Classifiers of COVID-19 Infection. *Cell Syst* **11**, 11–24.e4 (2020).
12. Dermitt, M., Peters-Clarke, T. M., Shishkova, E. & Meyer, J. G. Peptide Correlation Analysis (PeCorA) Reveals Differential Proteoform Regulation. *J. Proteome Res.* **20**, 1972–1980 (2021).
13. Kawahara, R. *et al.* Community evaluation of glycoproteomics informatics solutions reveals high-performance search strategies for serum glycopeptide analysis. *Nat. Methods* **18**, 1304–1316 (2021).
14. Bern, M., Kil, Y. J. & Becker, C. Byonic: advanced peptide and protein identification software. *Curr. Protoc. Bioinformatics* **Chapter 13**, Unit13.20 (2012).
15. Polasky, D. A., Yu, F., Teo, G. C. & Nesvizhskii, A. I. Fast and comprehensive N- and O-glycoproteomics analysis with MSFragger-Glyco. *Nat. Methods* **17**, 1125–1132 (2020).
16. Hoffmann, M. *et al.* The Fine Art of Destruction: A Guide to In-Depth Glycoproteomic Analyses-Exploiting the Diagnostic Potential of Fragment Ions. *Proteomics* **18**, e1800282 (2018).
17. Hoffmann, M., Marx, K., Reichl, U., Wuhrer, M. & Rapp, E. Site-specific O-Glycosylation Analysis of Human Blood Plasma Proteins. *Mol. Cell. Proteomics* **15**, 624–641 (2016).
18. Messner, C. B. *et al.* Ultra-fast proteomics with Scanning SWATH. *Nat. Biotechnol.* (2021) doi:10.1038/s41587-021-00860-4.